# Hysteretic evolution of ice rises and ice rumples in response to variations in sea level

A. Clara J. Henry[1,2,3], Reinhard Drews[2], Clemens Schannwell[1], and Vjeran Višnjević[2]

[1]Max Planck Institute for Meteorology, Bundesstr. 53, 20146 Hamburg, Germany
[2]Department of Geosciences, University of Tübingen, Schnarrenbergstr. 94-96, 72076 Tübingen, Germany
[3]International Max Planck Research School on Earth System Modelling, Max Planck Institute for Meteorology, Hamburg, Germany

**Correspondence:** Clara Henry (clara.henry@mpimet.mpg.de)

**Abstract.** Ice rises and ice rumples are locally grounded features found in coastal Antarctica and are surrounded by otherwise freely floating ice shelves. An ice rise has an independent flow regime, whereas the flow regime of an ice rumple conforms to that of the ice shelf and merely slows the flow of ice. In both cases, local highs in the bathymetry are in contact with the ice shelf from below, thereby regulating the large-scale ice flow, with implications for the upstream continental grounding line position. This buttressing effect, paired with the suitability of ice rises as a climate archive, necessitates a better understanding of the transition between ice rise and ice rumple, their evolution in response to a change in sea level, and their dynamic interaction with the surrounding ice shelf. We investigate this behaviour using a three-dimensional full Stokes ice flow model with idealised ice rises and ice rumples. The simulations span end-member basal friction scenarios of almost stagnant and fully sliding ice at the ice-bed interface. We analyse the coupling with the surrounding ice shelf by comparing the deviations between the non-local full Stokes surface velocities and the local shallow ice approximation (SIA). Deviations are generally high at the ice divides and small on the lee sides. On the stoss side, where ice rise and ice shelf have opposing flow directions, deviations can be significant. Differences are negligible in the absence of basal sliding where the corresponding steady state ice rise is larger and develops a fully independent flow regime that is well described by SIA. When sea level is increased and a transition from ice rise to ice rumple is approached, the divide migration is more abrupt the higher the basal friction. In each scenario, the transition occurs after the stoss side grounding line has moved over the bed high and is positioned on a retrograde slope. We identify a hysteretic response of ice rises and ice rumples to changes in sea level, with grounded area being larger in a sea level increase scenario than in a sea level decrease scenario. This hysteresis not only shows irreversibility following an equal increase and subsequent decrease in sea level, but also shows that the perturbation history is important when the ice rise or ice rumple geometry is not known. The initial grounded area needs to be carefully considered, as this will determine the formation of either an ice rise or an ice rumple, thereby causing different buttressing effects.

## 1 Introduction

Great progress in ice flow modelling has improved the physical representation of dynamical processes at the margins of the Antarctic ice sheet, but the transient evolution of the grounding line continues to be challenging, requiring high mesh resolution,

small time steps and advanced model physics (Schoof, 2007; Goldberg et al., 2009; Gudmundsson et al., 2012; Haseloff and Sergienko, 2018; Sergienko and Wingham, 2022). Moreover, the lack of past observational constraints and ice sheet model initialisation inconsistencies result in spin-up simulation geometries which differ from observations (Seroussi et al., 2019) and result in parameter choice uncertainty (Albrecht et al., 2020).

Ice rises and ice rumples are locally grounded features surrounded by floating ice shelves and play a dual role in this context. Firstly, ice rises and ice rumples regulate the flow of ice towards the ocean through their buttressing effect (Favier and Pattyn, 2015; Barletta et al., 2018; Reese et al., 2018; Still et al., 2019; Still and Hulbe, 2021; Schannwell et al., 2020) and influence the migration of the continental grounding line (Favier et al., 2012). Secondly, past adjustments in local ice shelf flow dynamics can be inferred from ice rises by investigating, for example, isochronal structure and the development of features such as Raymond arches within ice rises (Raymond, 1983; Martín et al., 2006; Gillet-Chaulet and Hindmarsh, 2011; Hindmarsh et al., 2011; Drews et al., 2013, 2015; Schannwell et al., 2019; Goel et al., 2020). The importance of ice rise formation and decay for continental ice sheet evolution (e.g., due to glacial isostatic uplift or changes in sea level) have been recognised in a number of scenarios and show the key role that ice rises play in large-scale grounding line migration patterns over glacial cycle timescales (Bindschadler et al., 1990, 2005; Barletta et al., 2018; Kingslake et al., 2018; Wearing and Kingslake, 2019).

In adopting terminology from Matsuoka et al. (2015), we identify ice rises as prominent grounded features with a distinct local radial flow regime, causing the flow of the surrounding ice shelves to divert either side of the feature. Ice rumples, however, generally form on less prominent bed highs and result in a predominantly unidirectional flow regime with the upstream ice shelf flowing over the bed anomaly. Ice rises and ice rumples are found all around the perimeter of the Antarctic Ice Sheet, but the mechanisms governing the transition from one flow regime to the other have not yet been investigated and influences of the surrounding ice shelves on the local flow regimes have not yet been quantified. In order to explore these questions, we use the three-dimensional, full Stokes model Elmer/Ice to simulate idealised ice rises and ice rumples under various basal friction scenarios and sea level perturbations.

To quantify non-local effects from the surrounding ice shelves, we compare the full Stokes solutions with the shallow ice approximation (Hutter, 1983; Greve and Blatter, 2009) and the Vialov profile (Vialov, 1958), which do not capture the stress transfer between ice shelf and ice rise. Furthermore, we investigate whether the locality of flow and basal sliding can be determined by examining the mismatch between the full Stokes ice thickness and the Vialov profile, an idealised solution for the ice geometry. Using sea level perturbations, we explore whether ice rises and ice rumples respond hysteretically and whether multiple steady states exist for a given set of boundary conditions by tracking the grounded area, upstream ice shelf velocity and dome position. Additionally, we investigate under which formation scenarios ice rumples reach a steady state and under which scenarios they are merely a transient feature during ice flow reorganisation.

## 2 Methods

Ice rises and ice rumples, and their surrounding ice shelves are investigated in steady state and transient scenarios using the three-dimensional full Stokes numerical model Elmer/Ice (Gagliardini et al., 2013).

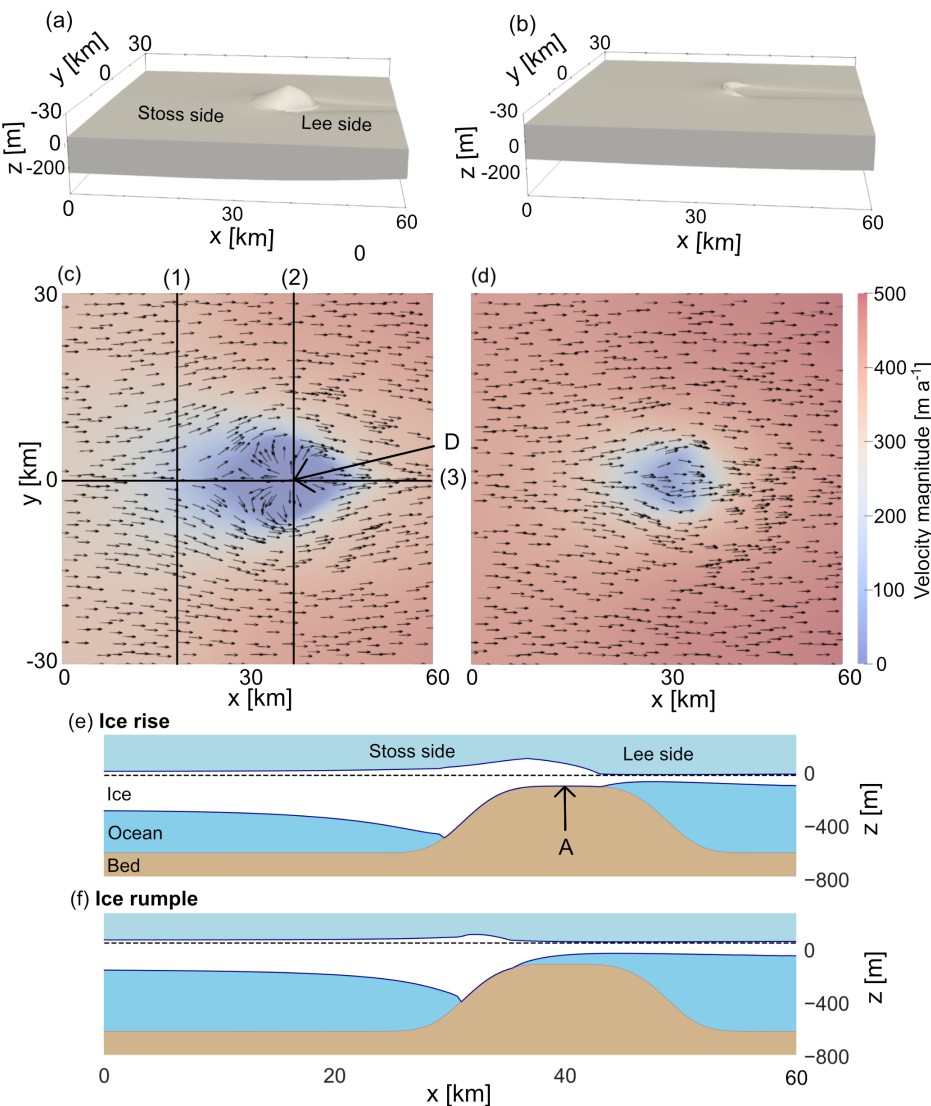

**Figure 1.** The $60 \times 60$ km model domain is shown in the case of (a) an ice rise and (b) an ice rumple. A corresponding bird's eye view in (c) and (d) show the surface velocity magnitude color coded and the ice flow direction with arrows. Corresponding along-flow cross-sections are shown in (e) and (f). Sea level is at an elevation of 0 m in the case of the ice rise and 80 m in the case of the ice rumple. In (c), (1), (2) and (3) indicate cross-sections used for analysis, and D is the ice rise dome. Both (2) and (3) are cross-sections through the ice rise dome. In (e), A marks the highest point of the bed anomaly. The $x$-direction corresponds with the along-flow direction, the $y$-direction corresponds with the across-flow direction and the $z$-direction corresponds with the elevation.

## 2.1 Governing equations

We adopt a coordinate system in which the predominant along-flow direction is aligned with the $x$-axis, the predominant across-flow direction is aligned with the $y$-axis and the $z$-direction marks elevation relative to sea level. The flow of ice is governed by the full Stokes equations,

$$\boldsymbol{\nabla} \cdot (\boldsymbol{\tau} - P\mathbf{I}) + \rho_i \mathbf{g} = 0, \tag{1}$$

where $\boldsymbol{\tau}$ is the deviatoric stress tensor, $P$ is the pressure, $\rho_i$ is the ice density and $\mathbf{g} = -g\hat{\mathbf{e}}_z$ is the gravitational acceleration. We assume the ice to be incompressible, and so, the mass conservation equation reduces to

$$\boldsymbol{\nabla} \cdot \mathbf{u} = 0. \tag{2}$$

The non-linear rheology of ice is modelled using Glen's flow law, which relates the deviatoric stress tensor, $\boldsymbol{\tau}$, to the strain rate tensor, $\dot{\boldsymbol{\epsilon}}$, as

$$\boldsymbol{\tau} = 2\eta\dot{\boldsymbol{\epsilon}}, \tag{3}$$

where the effective viscosity, $\eta$, is

$$\eta = \frac{1}{2} A^{-1/n} \dot{\epsilon}_e^{(1-n)/n}. \tag{4}$$

Here, $n$ is the Glen's flow law exponent, $A$ is a rheological parameter primarily dependent on ice temperature. Since we assume ice to be isothermal, A is set to a constant value in all simulations. The effective strain rate, $\dot{\epsilon}_e$, is calculated from the strain rate tensor, $\dot{\boldsymbol{\epsilon}}$, as

$$\dot{\epsilon}_e = \sqrt{\mathrm{tr}(\dot{\boldsymbol{\epsilon}}^2)/2}. \tag{5}$$

### 2.1.1 Boundary conditions

The upper surface, $z = z_s(x, y, t)$, evolves subject to

$$\left( \frac{\partial}{\partial t} + \mathbf{u} \cdot \boldsymbol{\nabla} \right)(z - z_s) = \dot{a}_s, \tag{6}$$

where $\dot{a}_s$ is the accumulation rate at the upper ice shelf surface. The lower surface, $z = z_b(x, y, t)$, evolves subject to

$$\left( \frac{\partial}{\partial t} + \mathbf{u} \cdot \boldsymbol{\nabla} \right)(z - z_b) = \dot{a}_b, \tag{7}$$

where $\dot{a}_b$ is the melt rate at the ice-shelf base. Furthermore, the grounded portion is constrained by the condition

$$b(x, y) \leq z_b(x, y, t) \leq z_s(x, y, t), \tag{8}$$

where $b(x,y)$ is the bed. The surface accumulation rate, $\dot{a}_s = 1.2$ m a$^{-1}$, reflects the comparatively high rates observed at some ice rises in the Dronning Maud Land area of East Antarctica (Drews et al., 2013). The basal melt rate, $\dot{a}_b$, beneath the ice shelf is defined as a function of ice thickness, $H$, based on the parameterisation used in Favier et al. (2016),

$$
\dot{a}_b = \begin{cases} 0, & \text{where ice is grounded, and} \\ \frac{1}{50} H^\alpha \tanh\left(\frac{|\mathbf{x}-\mathbf{x_g}|}{100}\right), & \text{where ice is floating,} \end{cases} \tag{9}
$$

where $\alpha$ is a tuning parameter and $|\mathbf{x}-\mathbf{x_g}|$ is the distance to the grounding line. During computation, $\mathbf{x}$ represents the position of the current node and $\mathbf{x_g}$ represents the position of the grounding line node closest to the current node. Here, $\mathbf{x}$ and $\mathbf{x_g}$ are expressed in km, $H$ is expressed in metres and $\dot{a}_b$ has units ma$^{-1}$.

A constant flux of $Q\big|_{x=0} = 5.4 \times 10^9$ m$^3$a$^{-1}$ into the domain is prescribed at the upstream boundary, corresponding to an initial velocity of $300$ ma$^{-1}$. Ice flows through a fixed calving front where ice is subject to sea pressure. At the lateral boundaries, a free-slip condition is applied and the flow velocity is subject to the Dirichlet boundary condition $\mathbf{u} \cdot \mathbf{n} = 0$, where $\mathbf{n}$ is the normal vector pointing outwards.

Ice in contact with the bed is subject to a non-linear Weertman-type friction law,

$$
\boldsymbol{\tau}_b = -C|\mathbf{u}_b|^{m-1}\boldsymbol{u}_b, \tag{10}
$$

where $\boldsymbol{\tau}_b$ is the basal shear stress, $C$ is a constant friction coefficient, $\boldsymbol{u}_b$ is the velocity tangential to the bed, and $m$ is the friction law exponent. The position of the grounding line at each time step is determined by solving a contact problem (Durand et al., 2009). The continuous *First Floating* Elmer/Ice grounding line numerical implementation is used (Gagliardini et al., 2016) and was chosen because a discontinuity in basal friction at the grounding line caused undesired numerical artefacts in the ice surface.

## 2.2 Idealised model domain setup

The evolution of ice rises and ice rumples is simulated in a $60 \times 60$ km domain (Fig. 1). A bed anomaly is introduced and allows isle-type ice rises and ice rumples to form. The bed takes the form $b(x,y) = b_0 + b_a$, where $b_0$ is a constant and $b_a$ is an anomaly with a flat top, defined as

$$
b_a(x,y) = M \exp\left\{ \frac{-((x-x_0)^2 + (y-y_0)^2)^2}{2\sigma^4} \right\}. \tag{11}
$$

The centre of the bed anomaly is located at $(x_0, y_0)$, $\sigma$ controls the horizontal extent and $M$ is the amplitude of the bed anomaly. The shape of the bed anomaly is broadly consistent with observations of ice rises, many of which have a plateau-shaped top that is near horizontal (e.g., Derwael Ice Rise (Drews et al., 2015) and ice rises in the Fimbul Ice Shelf (Goel et al., 2020)). All parameters used in the model are summarised in Table 1.

The ice thickness is initialised to $300$ m throughout the domain, resulting in a geometry that is predominantly floating with a small grounded area at the bed anomaly. To ensure adequate resolution at the grounding line and ice divide, the mesh is refined

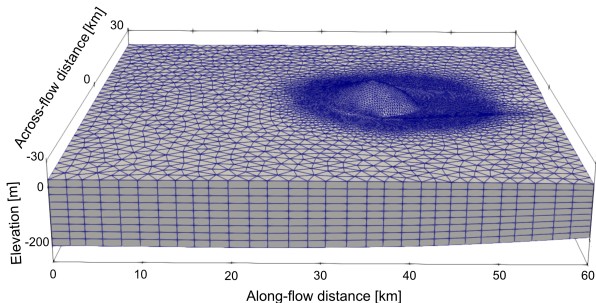

**Figure 2.** Shown is the mesh resolution of the mesh. In the horizontal, the mesh is unstructured and has a resolution of 350 m in the area surrounding the ice rise or ice rumple. The background resolution (of the surrounding ice shelf) is 2000 m. The mesh is extruded in the vertical with 10 layers. Note that the geometry is exaggerated by a factor of 30 in the vertical direction.

in the area encompassing the ice rise with a resolution of 350 m (Fig. 2). For this, we use the meshing software Mmg. This is in line with mesh resolution recommendations from other studies (Pattyn et al., 2013; Cornford et al., 2016), but is also the highest mesh resolution that is computationally feasible for the glacial-interglacial time scales considered here. To account for a possible migration of the ice rise, the radial extent of the area of high resolution is 5 km from the initial grounding line. In the remainder of the domain, a mesh resolution of 2000 m is used. The mesh is vertically extruded resulting in 10 layers spaced

equally apart and the horizontal mesh size is kept constant throughout the simulations.

### 2.3   Shallow ice approximation (SIA) comparison

The shallow ice approximation (Hutter, 1983; Greve and Blatter, 2009) describes the flow of ice in the absence of longitudinal and transverse stress gradients and is composed of the deformational velocity ($\mathbf{u}_d$) and basal sliding velocity ($\mathbf{u}_b$) so that the total velocity is $\mathbf{u} = \mathbf{u}_d + \mathbf{u}_b$. In SIA, only the vertical shear stress gradients are considered, so that the $x$-direction and

$y$-direction deformational components of the velocity take the form

$$\mathbf{u}_d = -2(\rho_i g)^n \boldsymbol{\nabla} z_s |\boldsymbol{\nabla} z_s|^{n-1} \int\limits_b^z A(T')(z_s - \bar{z})^n d\bar{z}. \tag{12}$$

We compare the velocity components only at the surface of the ice and also assume that temperature is constant, and so Eq. (12) reduces to

$$\mathbf{u}_d(x,y,z_s) = -\frac{2A(\rho_i g)^n}{n+1}(z_s - z_b)^{n+1} |\boldsymbol{\nabla} z_s|^{n-1} \boldsymbol{\nabla} z_s. \tag{13}$$

The $x$-direction and $y$-direction basal sliding components take the form

$$\mathbf{u}_b(x,y) = -C_b(\rho_i g(z_s - z_b))^{p-q} |\boldsymbol{\nabla} z_s|^{p-1} \boldsymbol{\nabla} z_s. \tag{14}$$

where $C_b$ is the basal friction coefficient and relates to the full Stokes basal friction coefficient, $C$, as follows:

$$C_b = \frac{N_b}{C^{1/m}} \tag{15}$$

**Table 1.** List of parameters used in the simulations

| Parameter | Symbol | Value | Unit |
|---|---|---|---|
| Rheological parameter | $A$ | $4.6 \times 10^{-25}$ | $\mathrm{Pa^{-3}\,s^{-1}}$ |
| Ice temperature | $T$ | $-15$ | $\mathrm{C^\circ}$ |
| Glen's exponent | $n$ | 3 | |
| Accumulation rate | $\dot{a}_s$ | 1.2 | $\mathrm{m\,a^{-1}}$ |
| Melt tuning parameter | $\alpha$ | 0.76 | |
| Glen's exponent | $n$ | 3 | |
| Basal friction exponent | $m$ | 1/3 | |
| Ocean density | $\rho_w$ | 1000 | $\mathrm{kg\,m^{-3}}$ |
| Ice density | $\rho_i$ | 900 | $\mathrm{kg\,m^{-3}}$ |
| Gravity | $g$ | 9.8 | $\mathrm{m\,s^{-2}}$ |
| Bed base | $b_0$ | $-580$ | m |
| Maximum bed height (above $b_0$) | $M$ | 500 | m |
| Bed anomaly extent parameter | $\sigma$ | 8 | km |
| Bed anomaly centre | $(x_0, y_0)$ | $(40, 0)$ | km |
| SIA basal drag exponents | $(p, q)$ | $(3, 1)$ | |

with

$$N_b = \rho_i g(z_s - z_b),$$
(16)

where $\mathbf{N}_b = N_b \mathbf{e}_z$ is the basal normal stress. In Eq. 14, $p$ and $q$ are chosen for consistency with the non-linear Weertman-type friction law described above.

## 2.4 Comparison with the Vialov approximation

The Vialov profile (Vialov, 1958) is an analytical solution for an ice sheet profile in the case of a non-slip, flat bed and constant accumulation. The flow in an ice rise is predominantly radial from a point divide and so we use a radial flux condition

$$\boldsymbol{\nabla} \cdot \boldsymbol{Q} = \frac{1}{R}\frac{\partial}{\partial R}(RQ_R) = \dot{a}_s,$$
(17)

assuming no azimuthal variance. Here, $\mathbf{Q} = Q_R \mathbf{e}_R$ denotes the vertically-integrated flux at a distance $R$ from the origin (located at the ice rise divide). The resulting ice geometry profile is of the form

$$h(R) = h_0 \left[ 1 - \left(\frac{R}{L}\right)^{\frac{n+1}{n}} \right]^{\frac{n}{2n+2}},$$
(18)

where

$$h_0 = 2^{\frac{n}{2n+2}} \left( \frac{\dot{a}_s}{2A_0} \right)^{\frac{1}{2n+2}} L^{\frac{n+1}{n}} \qquad (19)$$

and

$$A_0 = \frac{2A(\rho_i g)^n}{n+2}. \qquad (20)$$

$L$ is the horizontal distance from the ice rise divide to the grounding line, and both $h_0$ and $L$ are calculated from a reference
point on the surface of the full Stokes simulation output.

We compare only the lee profile of the ice rises to the Vialov profile as the bed is relatively flat in this area and we assume
that small changes in bed topography are negligible. The profiles are compared for a central cross-section from the divide,
extending in the along-flow direction into the ice shelf (Label (3) in Fig. 1c).

## 2.5 Design of transient simulations

To allow perturbation simulations to start from a steady state geometry, all simulations are run for 2000 years under con-
stant forcing. Simulations are performed for three different basal friction coefficients $C = 3.812 \times 10^6$, $C = 7.624 \times 10^6$ and
$C = 3.812 \times 10^8$ Pa m$^{-1/3}$ s$^{1/3}$, which we will refer to as *low*, *intermediate* and *high* friction scenarios, respectively. The
*intermediate* friction scenario has the same basal friction coefficient as that used in MISMIP (Pattyn et al., 2012) and in Favier
and Pattyn (2015), where an ice rise is also modelled. The *low* basal friction coefficient is close to the suggested value of
$3.16 \times 10^6$ Pa m$^{-1/3}$ s$^{1/3}$ in MISMIP+ (Cornford et al., 2020). The *high* basal friction scenario essentially excludes basal
sliding, mimicking ice frozen to the bed. For each basal friction coefficient, transient simulations with variable sea level are
performed (Fig. 3). In the *low* and *intermediate* basal friction scenarios, sea level is increased by $80$ m at a rate of $0.02$ ma$^{-1}$
over $4000$ years and then stays constant for another $2000$ years. Sea level is then decreased at a rate of $0.02$ ma$^{-1}$ back to
the initial level followed by a second phase of constant sea level for $2000$ years. A second cycle is performed for the *low*
basal friction scenario for a comparison with the first cycle. Branches of the *low* basal friction simulation are run to steady
state (equilibrium) at discrete intervals while keeping sea level fixed. We run these simulation branches in order to understand
how far from steady state the transient simulations are. The choice of sea level perturbation rate is in line with observations,
showing periods of sea level rise of up to $0.04$ m a$^{-1}$ during the last deglaciation (Deschamps et al., 2012). Furthermore, we
run branches of the simulation beyond the original sea level at the same sea level decrease rate of $0.02$ m a$^{-1}$.

In the *low* and *intermediate* scenarios, the ice rises transition to ice rumples at some stage during the sea level increase. In
the high friction scenario, no such transition occurs after a sea level increase of $80$ m. We therefore continue the increase of sea
level further at a constant rate of $0.02$ m a$^{-1}$ until the transition occurs. A reversal of the sea level perturbation is performed
from a height of $155$ m above the initial sea level.

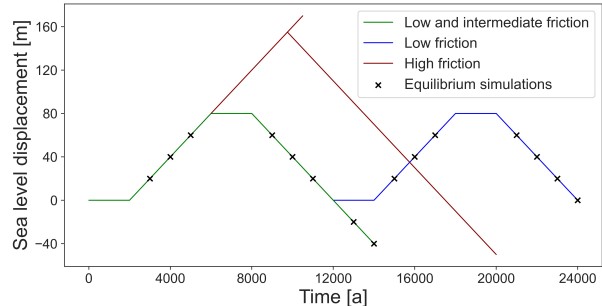

**Figure 3.** The change in sea level for the transient simulations. The *low* and *intermediate* scenarios follow the green curve. A second sea level increase and decrease cycle is performed for the low friction scenario (blue). Sea level is increased to $170$ m in the *high* friction scenario and a separate sea level decrease branch is simulated from $155$ m (red curve). Sea level is increased and decreased at rates of $\pm 0.02$ ma$^{-1}$. The crosses indicate points in the *low* friction scenario at which a steady state branch is started with constant sea level in order to compare to the transient simulation.

## 3 Results

### 3.1 Steady state analysis before sea level perturbation

After 2000 years of spin-up time, ice rises with a characteristic local flow regime develop in all three full Stokes scenarios (Fig. 4). From low to high friction, they vary in maximum thickness ($H_{max} = 213 - 468$ m), grounded area ($132 - 225$ km$^2$), and characteristic timescale ($t_c = 178 - 391$ a, defined as $t_c = H_{max}/\dot{a}_s$). The characteristic timescale is a metric that gives an indication of the rate of development of Raymond arches (Martín et al., 2009; Goel et al., 2020). The ice divide position in the *high* friction scenario has a stossward offset of $0.9$ km from the vertical symmetry axis of the bed protrusion. In the *intermediate* and *low* friction scenarios it is shifted stossward by $2.7$ and $3.3$ km, respectively. In all three cases, there is substantially more grounding on the stoss side of the bed protrusion than on the lee side.

Topographic and flow divides coincide in all three cases, and ice rise surface velocities are within tens of metres per year. There is negligible basal sliding in the high friction scenario (with average absolute velocities of roughly $0.5 \times 10^{-4}$ m a$^{-1}$ at the bed-ice interface), whereas basal sliding in the along-flow cross-section (Label (3) in Fig. 1c) accounts for 90 % and 98 % of the local mean horizontal velocities in the *intermediate* and *low* friction scenarios, respectively. The width of the lateral shear zones, here defined as the lateral distance from the grounding line along a cross-section (Label (2) in Fig. 1c) in which $v_x$ reaches 90 % of $v_x$ at the domain boundary, vary marginally from 10 to 11.3 km from the *low* to the *high* friction scenarios. Ice fluxes upstream of the protrusion are approximately equal, but mean velocities are 15 % slower and ice is about 15 % thicker in the *high* friction scenario compared with the *low* friction scenario.

All ice rises exhibit geometries and flow regimes which are comparable to observations. For example, the *high* friction scenario is comparable to Derwael Ice Rise, where previous studies have assumed no basal sliding a priori (e.g. Drews et al.

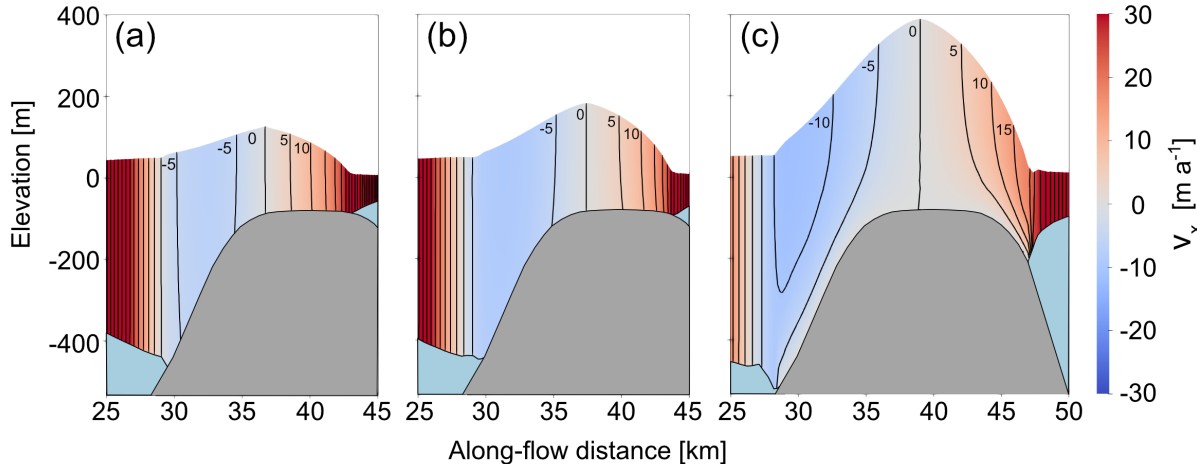

**Figure 4.** A cross-section of the ice rise in the along-flow direction for (a) the low basal friction, (b) the intermediate basal friction and (c) the high basal friction scenarios. The contours show lines of equal velocity (full Stokes) in the $x$-direction, i.e. in the along-flow direction.

(2015)). Basal sliding in the *low* and *intermediate* basal friction scenarios means these ice rises are more susceptible to transition into ice rumples when sea level is raised, as shown later.

The comparison of full Stokes surface velocities to SIA surface velocities on ice rises illustrates where the local flow assumptions are violated. Fig. 5 shows that all three basal friction scenarios have mismatches near the ice rise divides where longitudinal stress gradients are significant. The *high* basal friction scenario shows a good fit otherwise, as do the *low* and *intermediate* basal friction scenarios on the lee sides. However, for these cases, surface velocities differ more on the stoss sides of the ice rises (Fig. 6). In the *low* friction scenario, absolute deviations increase from 0-20 % in the vicinity of the divide (but not at the divide), to over 100 % closer to the grounding line. In the *intermediate* friction scenario, deviations are not quite as significant, but nonetheless reach a deviations of 100 %. In terms of ice thickness, the Vialov approximation captures the *high* friction scenario well despite the non-flat bed, while it significantly overestimates the *low* and *intermediate* basal friction scenarios in which basal sliding is dominant (Fig. 7).

## 3.2 Ice rise to ice rumple transitions triggered by sea level variation

To understand the response of ice rises and ice rumples with differing basal friction to sea level perturbation, we analyse the grounded area (Figs. 8a,b and 9a), dome migration (Fig. 10), lee side grounding line position and the upstream ice shelf velocities (Figs. 8c,d and 9b). The upstream ice shelf velocity is defined as the mean velocity of ice in the $x$-direction at $x = 20$ km, as marked by Label (1) in Fig. 1c. In terms of these metrics, the *low* and *intermediate* basal friction scenarios behave distinctly different than the *high* basal friction scenario. The former transition gradually to ice rumples if sea level is raised past a certain threshold and regrow into ice rises if sea level is reversed. The reversal is not symmetric and the respective steady state geometries depend on the history of their evolution (i.e. hysteresis). The *high* basal friction scenario, on the other hand,

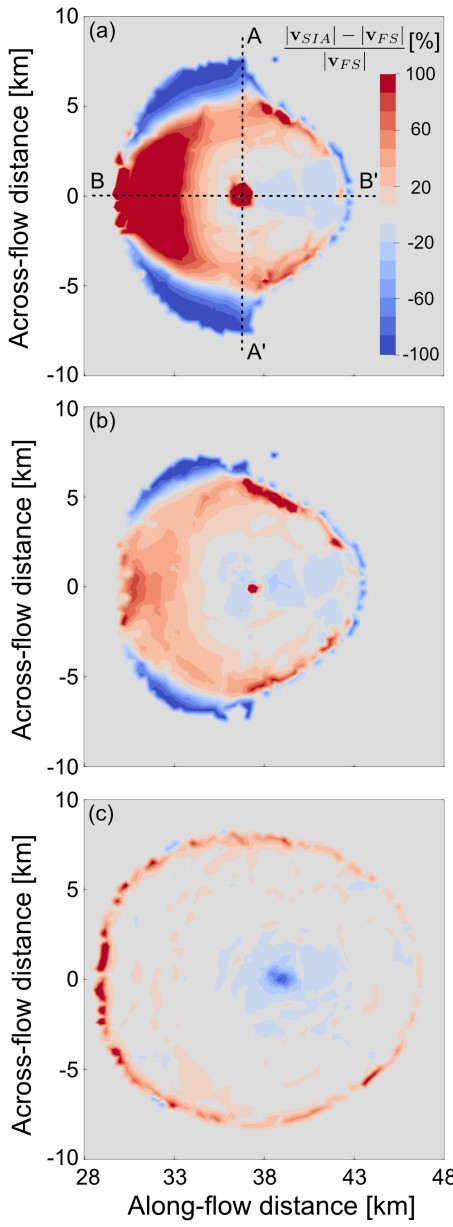

**Figure 5.** A bird's eye view of the grounded area corresponding to the steady states at $t = 2000$ years of the simulations with (a) a *low* basal friction, (b) an *intermediate* basal friction and (c) a *high* basal friction. In colour, the percentage difference is shown between the calculated SIA surface velocity magnitude and the full Stokes velocity magnitude.

requires a much larger sea level perturbation to trigger transition into an ice rumple. Once this transition is reached, an ice rumple forms but the system is unstable and the ice rumple ungrounds entirely. Details of these differing states are provided in the following.

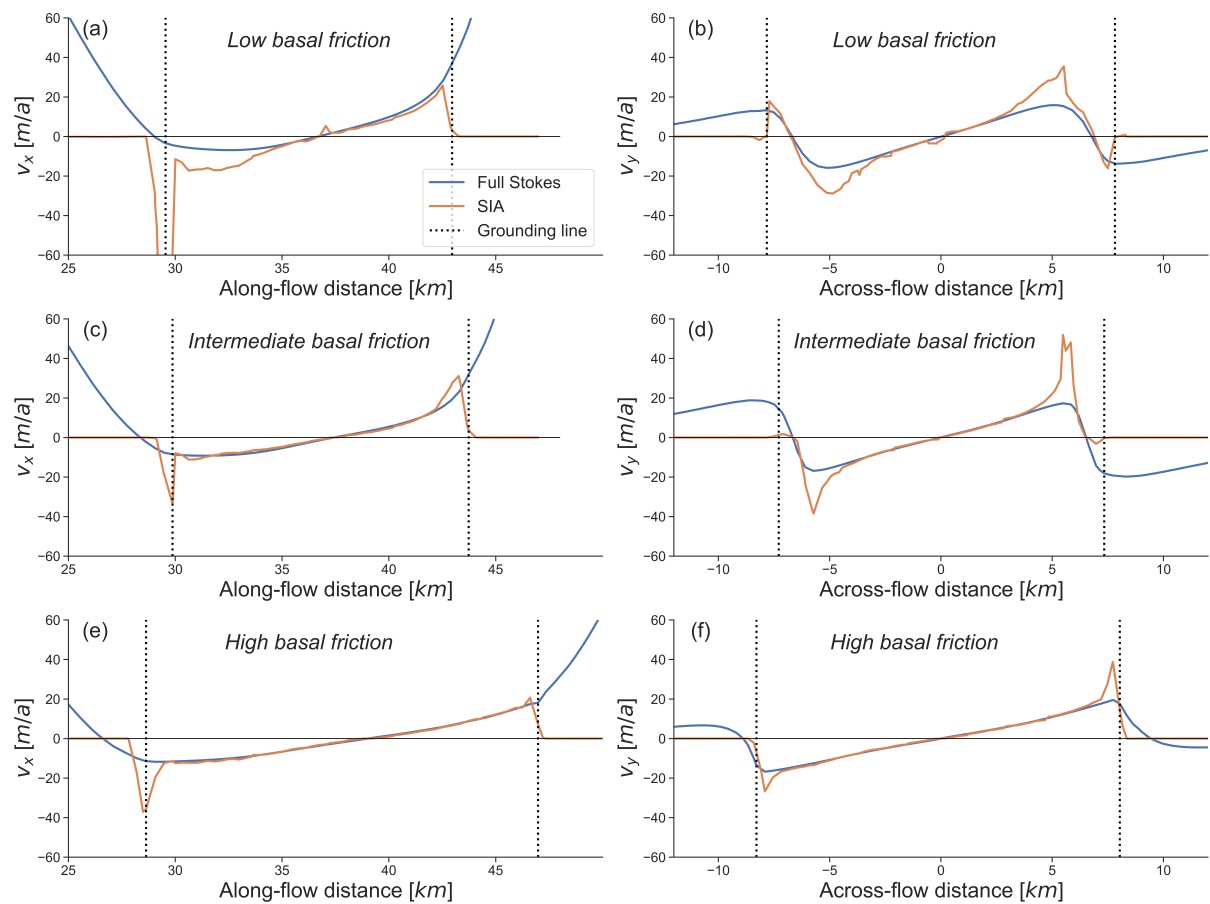

**Figure 6.** The full Stokes and SIA surface velocities at $t = 2000$ years in the along-flow direction ((a), (c) and (e)), and in the across-flow direction ((b), (d) and (f)), as indicated by the cross-sections A-A' and B-B' through the divide in Fig. 5. Figures (a) and (b) show the *low* basal friction scenario, (c) and (d) show the *intermediate* basal friction scenario, and (e) and (f) show the *high* basal friction scenario.

Before transitioning to an ice rumple, the dome position in the *low* friction scenario migrates linearly at a rate of $1.7 \text{ m a}^{-1}$ with increasing sea level (Fig. 10). The dome of the *intermediate* friction ice rise migrates first at a rate of $0.8 \text{ m a}^{-1}$ before increasing to a migration rate of $5.7 \text{ m a}^{-1}$ after a sea level increase of $29 \text{ m}$. The dome of the *high* basal friction ice rise exhibits a slow response to sea level displacement during the first $152 \text{ m}$ of sea level increase with a divide migration rate of $0.2 \text{ m a}^{-1}$, before increasing to a migration rate of $5.0 \text{ m a}^{-1}$.

After a sea level increase of $20 \text{ m}$ in the case of the *low* friction case, $30 \text{ m}$ in the case of the *intermediate* friction case, and $161 \text{ m}$ in the high friction case, the grounding line on the lee side of the ice rise migrates past the highest point of the bed anomaly (marked by A in Fig. 1e), and so, is located on a retrograde slope. A transition from ice rise to ice rumple occurs at a further sea level displacement of 30, 16 and 1 m after the grounding line has reached this point in the case of the *low*, *intermediate* and *high* basal friction scenarios, respectively.

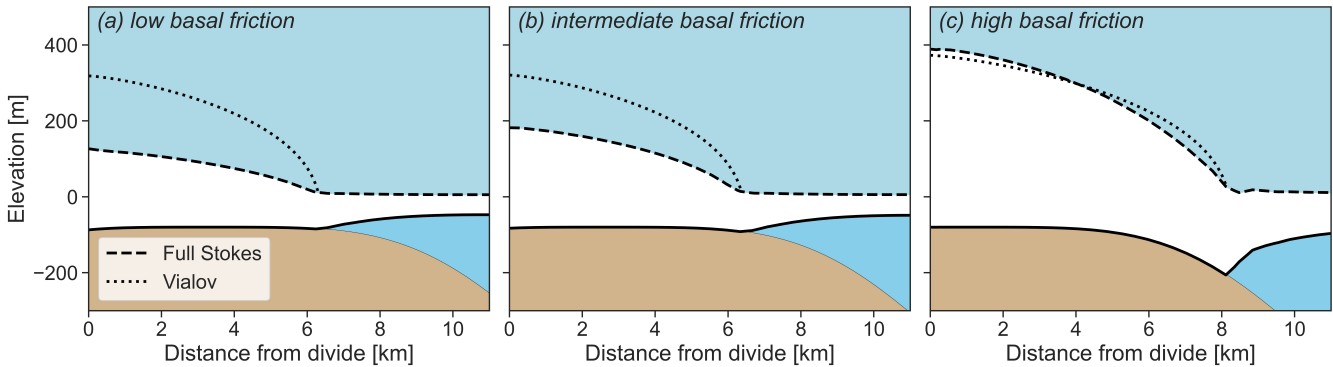

**Figure 7.** Cross-sections of the full Stokes simulations at $t = 2000$ years in the case of the (a) *low* basal friction coefficient, (b) the *intermediate* and (c) the *high* basal friction coefficient. Using a reference point on the ice surface at the grounding line, a Vialov profile is calculated and plotted.

A steady acceleration of the upstream ice shelf is seen in both the *low* and *intermediate* basal friction scenarios and there is no abrupt change once a transition from ice rise to ice rumple has occurred (Fig. 8). This is in contrast to the *high* basal friction scenario, where there is an abrupt change in the upstream ice shelf velocity as a transition from ice rise to ice rumple is approached.

After keeping the sea level constant for 2000 years at a sea level perturbation of 80 m, the *low* and *intermediate* basal friction ice rumples evolve to their respective steady states, with minimum velocities of 20 and 38 m a$^{-1}$ (Fig. 12). Reversal of the sea level perturbation then triggers an asymmetric reversal of the variables of interest described above, with grounded area and upstream ice shelf thickness increasing and upstream velocity decreasing. A transition from ice rumple to ice rise (Figs. 8 and A1) is observed when sea level is 21 and 19 m above the initial sea level in the *low* and *intermediate* basal friction scenarios, respectively (as opposed to displacements of 50 and 45 m for *low* and *intermediate* basal friction scenarios in the sea level increase scenarios, respectively). Once the original sea level is again reached, the ice rises in both the *low* and *intermediate* basal friction scenarios are smaller, with a smaller grounded area and a lesser buttressing effect on the upstream ice shelf (Figs. 8, A1 and 11). The upstream ice shelf in the case of the *low* basal friction scenario has a decrease in velocity of 18 m a$^{-1}$ whereas the ice shelf in the *intermediate* decreases in velocity by 25 m a$^{-1}$. A second cycle of sea level increase and decrease is performed for the *low* basal friction scenario starting from the steady states that emerged from the previous sea level perturbation cycle. The response of the grounded area and ice shelf velocity are calculated as described above and presented in Fig. 8. The hysteresis cycle is now closed, with the final steady state corresponding to the state before the last sea level perturbation cycle.

When sea level rise is halted in the *high* basal friction scenario prior to the unstable grounding line retreat (here at a sea level perturbation of 155 m), the ice rise volume and grounded area also recover asymmetrically resulting in two differing states for a given sea level displacement (Fig. 9).

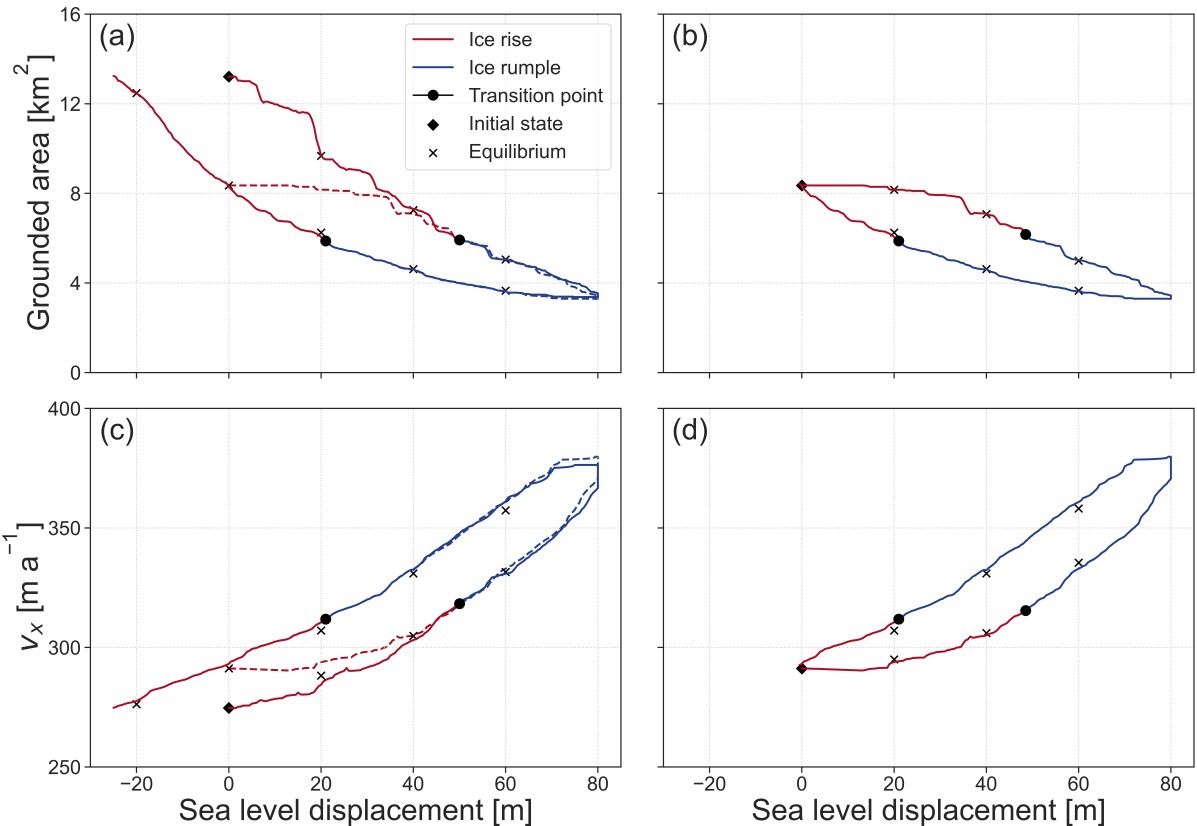

**Figure 8.** The response of grounded area and upstream ice shelf velocity to sea level perturbation in the case of the *low* basal friction. Panels (a) and (c) show the evolution for the first sea level increase and decrease cycle in blue and red. Panels (b) and (d) show the evolution for the second increase and decrease cycle. These curves are also plotted in panels (a) and (c) in with dashed red and blue lines for comparison. The crosses represent the results of steady state branches of the transient simulations at corresponding sea levels. The transition from ice rise to rumple and vice versa is represented by the black dots and a change in colour of the curve.

We investigate the migration of the stoss and lee side grounding lines of the ice rise and make a comparison with the grounding line position in the case of hydrostatic equilibrium (supplementary video). The maximum differences in position are 0.5 km on the stoss side and 0.4 km on the lee side, with mean differences of 0.2 km in both cases. During sea level increase, the hydrostatic grounding line positions have a delayed response in comparison with the Elmer/Ice grounding line. On the other 245 hand, during sea level decrease, the hydrostatic grounding lines have a more rapid response.

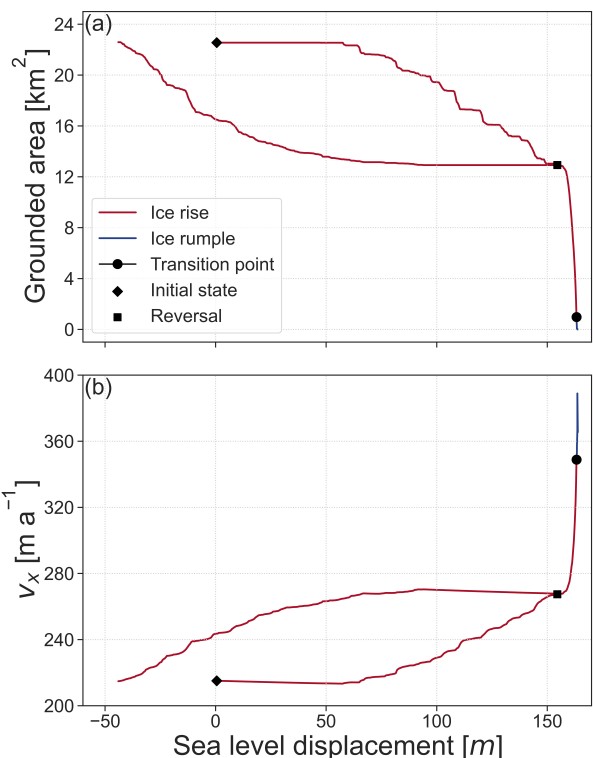

**Figure 9.** The response of grounded area and upstream ice shelf velocity to sea level perturbation in the case of the *high* basal friction. In (a), the grounded area is plotted against sea level displacement and in (b), the average velocity in the $x$-direction in a cross-section upstream of the ice rise (at 20 km from the influx boundary). Red indicates that the system exhibits a characteristic flow regime of an ice rise and blue indicates that of an ice rumple. The square indicates from where a reversal of the sea level perturbation is simulated.

## 4 Discussion

### 4.1 The influence of basal sliding on the geometry and transient behavior of ice rises

A number of previous studies have argued that basal sliding near ice rise divides is negligible because thermomechanically coupled models often predict ice significantly below freezing point at the ice-bed interface near the summits (Martín et al.,
2009; Drews et al., 2015; Goel et al., 2020) and because many ice rises exhibit isochronal features called Raymond arches which do not form if basal sliding is dominant (Pettit et al., 2003; Martín et al., 2009). However, *low* and *intermediate* scenarios can be relevant in areas where Holocene marine sedimentation results in basal sliding in areas which have regrounded (Pollard et al., 2016). Moreover, differences between observed and simulated Raymond arches under a frozen bed assumption may indicate a delay or suppression of arch growth due to past or present basal sliding (Kingslake et al., 2016).
The simulations show that ice rises can form in scenarios where basal sliding is significant. Surface velocities in the *low* and *intermediate* scenarios are within a few meters per years near the crests, similar to the predictions in the *high* friction scenario

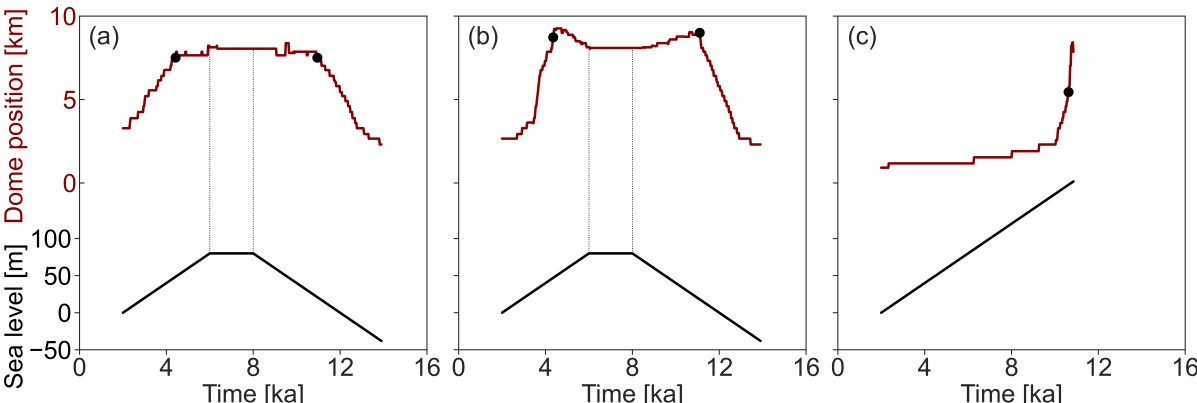

**Figure 10.** The response of the dome position to a raising and lowering of sea level in the case of (a) the low, (b) the intermediate and (c) the high basal friction coefficients.

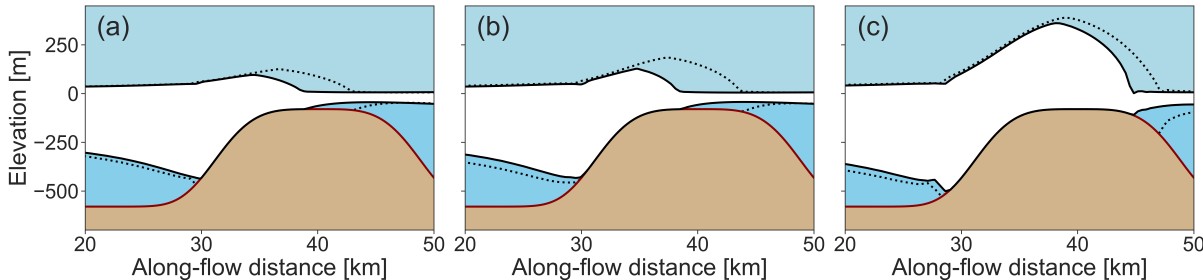

**Figure 11.** The figures show a cross-section of the ice rises in the along-flow direction for (a) the *low*, (b) the *intermediate* and (c) the *high* basal friction scenario. The dotted lines show the geometry of the ice rises before sea level perturbation and the solid lines show the geometry after a full cycle of sea level increase and decrease.

(Fig. 5). In this regard, surface velocities alone are a poor indicator for the presence or absence of basal sliding on ice rises. However, the geometries between the three scenarios differ significantly, and only the high friction scenario can be adequately approximated with the Vialov profile whereas the *low* and *intermediate* scenarios exhibit significant misfits (Fig. 8). This means
that a simple fit with a Vialov profile can serve as a first order metric for absence or existing of basal sliding for specific ice rises. This is important, as the degree of basal sliding in the vicinity of the grounding line determines the local ice flow and the ice rise's transient behaviour in response to sea level perturbation. When comparing the grounding line positions of the full Stokes model and the hydrostatic grounding line position, we find that differences are small. However, over the millennial timescales considered here, together with the compounding effect of the small errors in grounding line position at each time
step, it is possible that a hydrostatic assumption may result in differing ice rise and ice rumple geometries as well as a differing transition point.

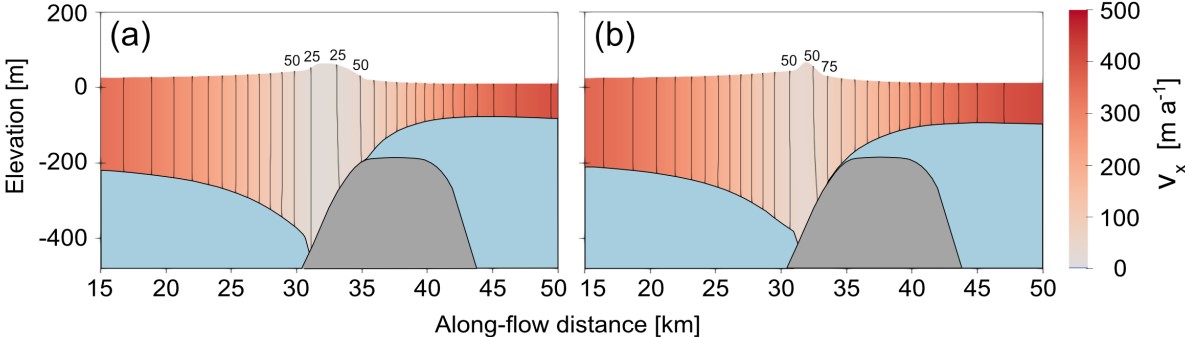

**Figure 12.** An along-flow cross-section of the ice rumple at $t = 8000$ years in the case of (a) the low basal friction and (b) the intermediate basal friction. The contours show lines of equal velocity in the $x$-direction and are spaced $25 \text{ m a}^{-1}$ apart.

Many ice rises are fully surrounded by ice shelves and the extent to which isle-type ice rise velocities are affected by longitudinal and shear stresses transferred from the upstream ice shelf is not fully clear. This effect is analysed here using the differences between the non-local full Stokes simulations and the fully local SIA. The flow regime in the *high* friction scenario is, to a large extent, independent of the surrounding ice shelf. In the *low* and *intermediate* basal friction scenarios, however, the differences between full Stokes and SIA are greater, and are especially evident on the stoss side of the ice rise. The greater velocity differences in the lower friction scenarios show that these ice rises are influenced more by the stresses in the surrounding ice shelf. Implications for the presence or absence of a fully local flow regime are twofold: (1) if basal sliding is negligible even in areas close to the grounding zone, then SIA is an appropriate modelling framework, for example, when investigating the surface accumulation history using inverse methods (Callens et al., 2016), and (2) the basal boundary condition determines an ice rise's response to sea level perturbation.

The *low* and *intermediate* friction scenarios respond immediately to a rising sea level, with a retreat of the leeward grounding line accompanied by a stossward migration of the dome position. The ice rises progressively thin and eventually transition into ice rumples. There is no significant threshold behaviour between these two states and once the sea level increase is halted, the system converges to a steady state ice rumple with the lee side grounding line located on the retrograde slope at the edge of the basal plateau. The summits are a few tens of meters above the ice shelf surface and the overall geometry is consistent with, for example, the ice rumple located in the Roi Baudouin Ice Shelf (Fig. 13). The minimum overriding velocities of $20 \text{ m a}^{-1}$ are, however, significantly faster than the example observed at the Roi Baudouin Ice Shelf where the ice is effectively stagnant (Berger et al., 2016). The smooth transition of the *low* and *intermediate* friction ice rises into ice rumples reflects their strong coupling to the surrounding ice shelf, highlighted previously. From a larger scale perspective there are no critical differences between ice rises and ice rumples in those scenarios other than the switch from a local to an overriding flow regime.

Conversely, the *high* friction case only transitions to an ice rumple for sea level perturbations that are greater than what is expected in a glacial-interglacial cycle. In fact, there is no noticeable change in grounded area even for a sea level displacement

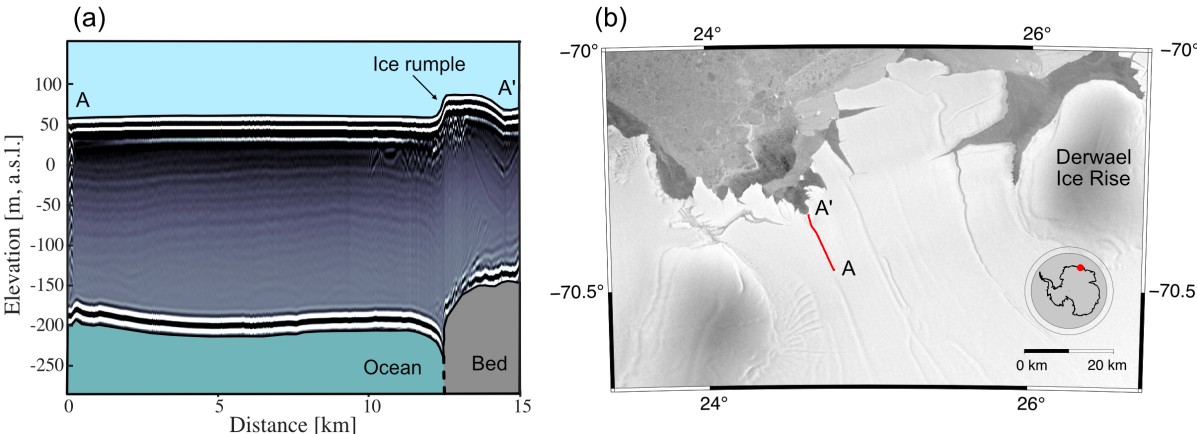

**Figure 13.** An along-flow ground-based radargram (Drews, 2019) showing an ice rumple in the Roi Baudouin Ice Shelf, East Antarctica is shown in (a). The flow of ice is from left (A) to right (A'). In (b), the location of the radargram (A-A') is shown (Jezek, 2003).

of 50 m. This stability is in line with, for example, ice promontories at the Ekström Ice Shelf which show a comparatively weak
response to the thinning of their surrounding ice shelves (Schannwell et al., 2019). Grounding line retreat rates for higher sea level displacements then remain moderate as long as the leeward side remains grounded on a prograde slope. On a retrograde slope the ice rise becomes unstable and complete ungrounding occurs. We therefore conclude that after a transition from ice rise, there is a threshold basal friction beyond which a steady state ice rumple cannot form.

Interestingly, the *low* friction ice rumple exhibits lower minimum velocities than the *intermediate* friction ice rumple, most
likely due to a greater grounded area (Fig. 12) and it is worth investigating whether inverse techniques used to predict the basal friction coefficient beneath pinning points produce results which remain valid regardless of horizontal resolution applied.

The required sea level perturbation for ungrounding clearly depends on the elevation below sea level of the bed protrusion, but the scenarios shown here with a maximum bed elevation of 80 m below sea level have many real world counterparts (e.g., Kupol Moskovskij, Kupol Coilkovskogo, Leningrad Ice Rise, Djupranen Ice Rise (Goel et al., 2020), Derwael Ice Rise
(Drews et al., 2015)). Our study suggests that features with a high basal friction have been and will remain stable local flow features even for comparatively large sea level perturbations. Moreover, it shows that ice rumples with comparatively low surface velocities as in the example provided in Fig. 13, are very unlikely a result of a deglaciated ice rise. An area that requires more investigation is the case of ice rises which do not conform to the plateau-shaped bed topography as prescribed here. The unstable retreat predicted in the high basal friction scenario suggests that ice rises located on retrograde slopes are critically
less stable for an equal amount of sea level displacement.

## 4.2 The hysteretic behaviour of ice rises over glacial cycles

In all basal friction scenarios, there are two differing ice rises for a given sea level (Figs. 8, 9). These pairs differ in the basal melt rate applied (which is thickness dependent) and in the grounded area. Each pair corresponds to a low and a high buttressing case for which the averaged upstream ice velocity is used as a proxy (Figs. 8c,d, 9b and Fig. A1b in the Appendix).

There is a difference in the individual pairs, with the grounded area being larger in the sea level increase scenario than in the sea level decrease scenario. In all cases, the pairs occupy virtually the same region on the obstacle's stoss side, but the extent of grounding on the plateau differs (Fig. 11). The thickness and slopes at the respective grounding lines are comparable, and therefore differences in basal melt (as parameterised in Eq. 9) are small, with differences of only 3.5, 3.0 and 2.4 % in the *low*, *intermediate* and *high* friction scenarios, respectively. The dynamic differences therefore stem mostly from the differing grounded areas that result in a differing form drag (Still et al., 2019) and consequently a differing net resistance to the upstream ice shelf.

A self-stabilising feedback occurs, with divide migration opposing grounding line retreat in a sea level increase scenario. The ice rise height reduces and the divide migrates stossward during lee side grounding line retreat. Because the divide moves stossward, the area of accumulation adjacent to the divide on the lee side of the ice rise increases. The increased accumulation area promotes an increased flux across the grounding line, opposing grounding line retreat. Analogously, sea level decrease results in leeward divide migration. The resulting reduction in accumulation area adjacent to the divide on the lee side of the ice rise opposes grounding line advance. The existence of negative feedback mechanisms in both the sea level increase and decrease scenario result in hysteretic behaviour (Figs. 8, 9, A1).

Another mechanism that plays a role is the sensitivity of the grounding line to bed shape, with hysteretic behaviour occurring due to the positioning of retrograde and prograde slope segments (Schoof, 2007; Pattyn et al., 2012; Haseloff and Sergienko, 2018; Sergienko and Wingham, 2022). In our study, we also observe grounding line migration patterns linked to the shape of the three-dimensional bed protrusion. Consequently, it matters how the ice rise and ice rumple geometries are initialised to begin with.

Although in our study, we have used a constant surface accumulation, we would expect orographic precipitation to enhance the hysteretic behaviour. In future work it is worth investigating whether effects such as an increased melt rate also produce an hysteretic response in ice rises and ice rumples. Given that the grounded area and basal sliding determine the ice rise evolution, future simulations should include a more informative guess of the basal friction coefficients guided by, for example, seismic studies determining the bed properties (Smith et al., 2015). Inversion of the basal friction parameters from a thermomechanically coupled full Stokes model (Schannwell et al., 2019, 2020) does provide some information in this regard, but also contains lumped uncertainties, e.g., from ice rheology and uncertain boundary conditions. Another process not considered here is changes in the bed protrusion through glacial isostatic adjustment (Kingslake et al., 2018; Wearing and Kingslake, 2019).

The existence of multiple steady states means that the grounding lines of ice rises and ice rumples observed today are dependent on the local ice flow history during the last glacial cycle. Inversely, the dynamics and buttressing effect of ice rises and ice rumples are dependent on the initial geometry prescribed, which is typically unknown. The degree of buttressing is of

importance for determining the stability and evolution of the continental grounding line (Favier and Pattyn, 2015; Reese et al., 2018). The representation of ice shelves has been identified as a key cause of continental-scale model spread (Seroussi et al., 2019) and a precise representation of ice rises and ice rumples would reduce spin-up and projection uncertainties.

We have shown that the difference between the simulated grounding line and the hydrostatic equilibrium grounding line is small at each time step. This small error may, however, lead to an error propogation during transient simulation leading to
inaccurate grounding line migration if a hydrostatic equilibrium assumption is used.

## 5   Conclusions

We examined the effect of basal friction and sea level variation on the evolution of ice rises and ice rumples using idealised simulations including the surrounding ice shelves. In a high basal friction scenario, there is negligible mismatch when comparing simulated steady state full Stokes velocities with steady state SIA velocities, whereas in a low basal friction scenario the
mismatch is larger due to stronger mechanical coupling to the surrounding ice shelf. The locality of the ice flow and the degree of basal sliding can be diagnosed by examining the (mis-)fit of a Vialov profile to the observed thickness profile. In response to an increasing sea level, a transition from ice rise to ice rumple occurs. Steady state ice rumples form in the low basal friction scenarios whereas the ice rumple in the high friction scenario is ephemeral and ungrounds rapidly. The higher friction ice rise, on the other hand, is largely unresponsive to sea level variations, requiring more than double the sea level rise to trigger the
transition compared to the lower friction scenarios.

All basal friction scenarios show self-stabilising, hysteretic behaviour, with grounded area and upstream ice shelf buttressing dependent on the evolution history. As a consequence of this behaviour, we identify the importance of perturbation history for the formation of the correct feature. Although in our study, we have concentrated only on the response of ice rises to sea level perturbation, further processes such as an increase in basal melt are also likely to result in hysteretic and potentially irreversible
behaviour in ice shelf buttressing upstream of ice rises.

*Code availability.* The code used to run the simulations and the post-processing code can be found at https://doi.org/10.5281/zenodo. 6355565. The Elmer version is Version: 8.4 (Rev: 1c584234)

*Video supplement.* A supplementary video is provided, showing the evolution of an ice rise in response to sea level perturbation as well as the position of the grounding line if the system were in hydrostatic equilibrium.

 **Appendix A:  The response of the grounded area and ice shelf velocity to sea level perturbation in the *intermediate*
basal friction scenario.**

Presented in Fig. A1 is the response of the grounded area and upstream velocity to sea level perturbation in the case of the
*intermediate* ($C = 7.624 \times 10^6$ Pa m$^{-1/3}$ s$^{1/3}$) basal friction scenario. The transition from ice rise to ice rumple occurs at a sea
level displacement of 19 m and the transition from ice rumple to ice rise occurs at a sea level displacement of 45 m, compared
with 21 m and 50 m, respectively, in the *low* basal friction scenario). Interestingly, the grounded area of the ice rumple follows
a rather linear path in the *intermediate* basal friction scenario compared with the *low* basal friction scenario.

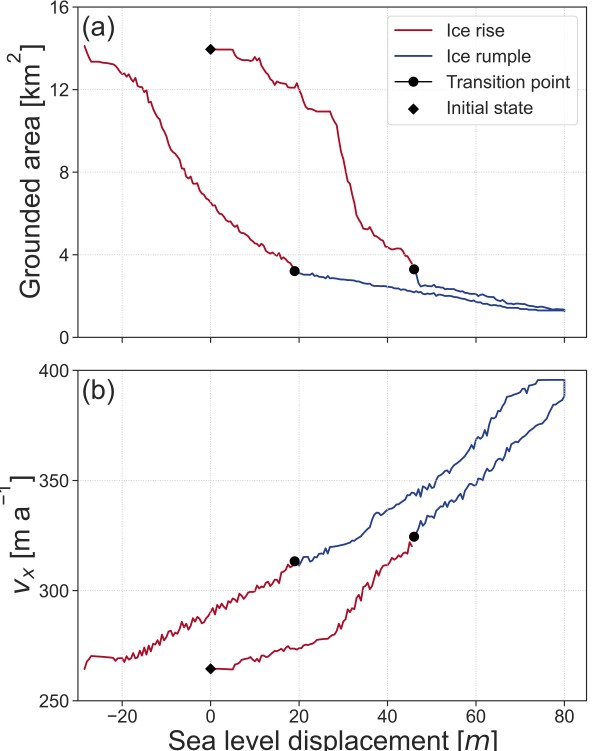

**Figure A1.** The response of the grounded area and ice shelf velocity to sea level perturbation in the *intermediate* friction scenario. In (a),
the grounded area is plotted against sea level displacement and in (b), the average velocity in the $x$-direction in a cross-section upstream of
the ice rise (at 20 km from the influx boundary). Red indicates that the system exhibits a characteristic flow regime of an ice rise and blue
indicates that of an ice rumple.

**Appendix B:  Comparison between the *First Floating* and *Discontinuous* grounding line numerical implementations**

In the case of the *low* basal friction scenario, we have run equivalent simulations using a differing grounding line numerical
implementation, namely the *Discontinuous* method (Fig. B1). At the grounding line, basal friction is applied if the other two

nodes in the element are also grounded and a free-slip condition is applied if the other two nodes are ungrounded. The *First Floating* numerical implementation, however, assumes a free-slip condition at the grounding line and a linear reduction in basal friction between it and the upstream node is applied. Although the *Discontinuous* numerical implementation has been shown to have the least dependence on mesh resolution, it can be argued that the *First Floating* is more plausible physically, with effective pressure disappearing at the grounding line (Gagliardini et al., 2016). The simulations show that regardless of

the numerical implementation, hysteresis occurs.

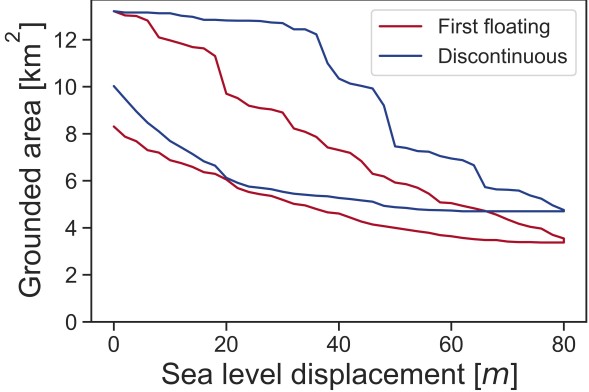

**Figure B1.** Shown is the response of the grounded area in the low friction case of the *First Floating* (red) and *Discontinuous* (blue) Elmer/Ice numerical grounding line implementations.

*Author contributions.* C. Henry, C. Schannwell, and R. Drews conceived the idea for the study and designed the experiments. C. Henry performed the simulations and analysis. The manuscript was written by C. Henry with contributions from all authors.

*Competing interests.* R. Drews is an editor for The Cryosphere.

*Acknowledgements.* C. Henry was supported by the Deutsche Forschungsgemeinschaft (DFG) in the framework of the priority programme

1158 "Antarctic Research with comparative investigations in Arctic ice areas" by a grant SCHA 2139/1-1. C. Schannwell was supported by the German Federal Ministry of Education and Research (BMBF) as a Research for Sustainability initiative (FONA) through the PalMod project under the grant number 01LP1915C. R. Drews and V. Višnjević were supported by an Emmy Noether Grant of the Deutsche Forschungsgemeinschaft (DR 822/3-1). This work used resources of the Deutsches Klimarechenzentrum (DKRZ) granted by its Scientific Steering Committee (WLA) under project ID bm1164. The authors gratefully acknowledge the Gauss Centre for Supercomputing e.V.

(www.gauss-centre.eu) for funding this project by providing computing time on the GCS Supercomputer SuperMUC-NG at Leibniz Supercomputing Centre (www.lrz.de). Data for Fig. 13 were collected with the support of the InBev Baillet Latour Antarctica Fellowship with logistic support from the International Polar Foundation.

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
