# Peer review of "Hysteretic evolution of ice rises and ice rumples in response to variations in sea level"

_EGUsphere, 2022_

## Referee Comment (RC1)

**General comments**

The paper 'Hysteretic evolution of ice rises and ice rumples in response to variations in sea level' by Clara Henry, Reinhard Drews, Clemens Schannwell and Vjeran Višnjević is a modeling study making use of the Finite Element code Elmer/Ice in order to investigate, from synthetic three-dimensional scenarios, the stability of ice rises and ice rumples, as well as the dynamical transition from one flow regime to the other depending on the amount of friction at the ice/bed interface. To this end, starting from an initial steady state corresponding to an ice rise situation, perturbation experiments consisting in cycles of sea level rise and decrease are run solving the full-Stokes set of equations. Obtained initial steady surface velocities on the grounded part are compared to their Shallow Ice Approximation (SIA) counterparts in order to quantify the importance of longitudinal stresses transmitted from the surrounding ice shelf to the grounded ice. Unsurprisingly deviations are significant when basal friction is low, whereas they become negligible in the high basal friction scenario. The transient simulations show that an increase of sea-level induces a transition from an ice rise flow regime to an ice rumple regime in all friction scenarios. However, in the high friction scenario, much higher sea level increase is required than in the other scenarios to switch from the ice rise to the ice rumple regime, and the latter is unstable (i.e. complete ungrounding rapidly occurs). Interestingly, the sea level decrease experiments bring to light a hysteretic response of grounded ice, with the grounded area and induced buttressing effect being systematically lower than in the sea level increase phases when sea level is decreased back to its initial level. Conclusions are then drawn regarding the initialisation of ice flow models as well as the inversion of basal friction parameters.

Overall, the paper is well-written, the proposed methodology is rigorous, the experiments are well-designed, the figures are mostly clear and relevant, the supplementary video is very illustrative, and the conclusions regarding the stability of ice rises, as well as the highlighted hysteretic behavior in response to sea-level changes and associated irreversibility are significant for improvement of the accuracy of sea level rise projections. Therefore, I think the paper ought to be published and I have only a few minor modifications/comments to propose.

My main point regards the logical link that is made between the hysteretic response of the ice rise to sea level rise and the requirement for careful model initialisation (e.g. 1.18 or 1338-339). I am not completely sure that this association really holds. Don't get me wrong, I totally agree that careful initialisation of models is of prime importance when running transient simulations of the future evolution of ice sheets/shelves. I also agree that "the dynamics and buttressing effect of ice rises and ice rumples are dependent on the initial geometry prescribed, which is typically unknown" (l.319-320). I see clearly the link between the hysteretic behavior and some form of irreversibility: if the system is forced with a given perturbation from a given initial steady state, it does not come back to the same steady state when the perturbation is removed. However, it does not necessarily mean it will behave dramatically differently if you start from a slightly different initial stead, as long as the perturbation pattern is similar (i.e. in your case, sea level increase OR sea level decrease). A good illustration of this point is the system starts for this second cycle is different from the one of the first cycle, and yet the dynamical evolution of the grounded area and buttressing effect become relatively rapidly similar to that of the first cycle (dotted lines are 'rapidly' superimposed to solid lines in Figs. 8a-c). Once again, I have the feeling that it is more the history of the perturbation (are we in a sea level increase or decrease phase?) that is of importance rather than the initial state.

Another point regards the presentation of the SIA model (Sect. 2.3 and 2.4). It seems to me that it is largely inspired from Greve and Blatter (2009), and some notations become inconsistent with the ones that were used to introduce the full-Stokes model in Sect 2.1. See specifics comments.

Finally, there are a few points that, in my opinion, lack of clarity. First, I would write straight away in the abstract that you are running synthetic experiments and not dealing with real-world applications. Second, the fact that the comparison between the full-Stokes and SIA surface velocities is done for the initial steady states only would benefit to be stated more clearly in the text. Finally, although it is never clearly mentioned in the text (unless I missed something), it seems from Figs.3-8-9-10 that the sea level decrease experiments are continued after the initial 0 m level has been recovered. If this is true, this would deserve some explanation in the text.

Below, I list some specific comments.

**Specific comments**

P2 L30: 'control'  $\rightarrow$  Is that not too strong ? What about 'influence' or 'affect' ?

P2 L46-47: 'simpler ice-flow approximations'  $\rightarrow$  shouldn't it be singular ?

P4 L62: I think there should be a minus sign in front of g as the vertical unit vector is pointing upward.

P4 L64: The 0 should not be bold as the divergence of a vector is a scalar.

P4 L76: Here, the vector **u** represents the velocities at the surface/base, which should be precised. There is an additional constraint that you did not mention and that is of importance for the grounded part, i.e.  $b(x, y) \leq z_b(x, y, t) \leq z_s(x, y, t)$  where b represents the bed elevation.

P5 L82: How do you tune this 'tuning parameter' ?

P5 L82: 'the closest'  $\rightarrow$  what do you mean by 'the closest' ?

P5 L85: 'no friction'  $\rightarrow$  I find more natural to speak about 'free-slip', but it is a matter of choice.

P5 L91: Why did you choose the 'First Floating' implementation ? You are dealing with Weertman friction law with constant friction coefficients, which induces a discontinuity of friction at the GL on the mathematical level. The proper way to represent this discontinuity in the FEM numerical framework of Elmer is to use the 'Discontinuous' implementation. Using the 'First Floating' implementation, you artificially impose a linear decrease of friction over all the last grounded elements due to the interpolation of friction parameters at the integration points from their nodal values through the FEM basis functions. This choice could be justified by the fact that, on the physical level, one would expect a smooth transition of friction from its value on the grounded part to zero at the GL. If that was your reasoning when choosing the FF implementation, then it would be worth to state it clearly in the text.

P6 L103: According to Gagliardini and others (2016), with such a resolution of 350 m, the GL behavior keeps being sensitive to the numerical treatment of friction at the GL (i.e. First Floating, Last Grounded or Discontinuous implementations). This is an additional reason to give a clear justification for your choice of the FF implementation.

P6 L105: It would be worth precising the order of magnitude of considered time scales here, even if it becomes clearer later in the text.

P6 Eq10: It is strange that you switch from  $z_s$  to h and from  $z_b$  to b to denote the surface/base elevations. It feels like you have taken Eq. (5.84) of Greve and Blatter (2009) without adapting the notation to your own work.

P6 Eq11: Same remark as above, plus I don't understand the purpose of the subscript h for the gradient operator. First, it does not appear in Eq. (10). Second, h being the surface elevation (your  $z_s$ ), it does obviously not depend on z so that the gradient operator does only consist in the x and y components. In addition, in the first occurence of the gradient operator in Eq. (11), the subscript is not at the proper position.

P6 Eq12: Here again, the formulation of Greve and Blatter (2009) would require some adaptations to your own case. More precisely, this expression is not directly consistent with the way the basal shear stress is defined in Eq. (8). In Greve and Blatter (2009), the effect of the normal stress  $N_b$  on the basal shear stress is explicited in the formulation of the friction law (5.35), while most of the time this effect is hidden in the friction parameter C of the Weertman law. I know that Eq. (13) is intended to establish the consistency between the two formulations, but in that case I think there should be a minus sign in Eq. (13) (or in Eq. (8), but you have to choose), and the exponents p and q that appear in Eq. (12) would need to be defined somewhere. In your case q = 1 and p = 1/m. I think it would be cleaner to have Eq. (12) formulated directly consistently with Eq. (8).

P6 Eq13: See comment above.

P6 Eq17: Here, the expression of  $h_0$  does not correspond to the one in Eq. (5.117) of Greve and Blatter (2009). Could you double-check ? P8 Fig. 3: The Fig. is not consistent with the main text of Sect. 2.5: the decrease of sea level seems to continue further below the initial 0 m level (green and red lines), while the text says that "Sea level is then decreased at a rate of  $0.02 \text{ m a}^{-1}$  back to the initial level followed by a second phase of constant sea level for 2000 years."

P8 L148: "with ice being frozen to the bed"  $\rightarrow$  "mimicking ice frozen to the ground" ?

P9 L162: "characteristic timescale"  $\rightarrow$  I have a hard time to figure out what does this quantity represents physically. Maybe, it requires more explanation or maybe you can simply drop it as it is not used anywhere else later on.

P9 L189-190:"the upstream ice shelf velocities"  $\rightarrow$  it is not very clear what velocities exactly you are referring to, and how you can get a single value of velocity. You give a brief explanation in the caption of Fig. 9, but some precisions would be required here too.

P10 Fig. 4: You should precise that the velocities are the full-Stokes ones (are they ?).

P11 Fig. 5: What is represented exactly ? Is is  $|v_{stokes} - v_{SIA}|/|v_{stokes}|$ ? If yes, wouldn't it be more interesting to represent  $|(v_{stokes}| - |v_{SIA}|)/|v_{stokes}|$ , so that we could tell when the full-Stokes surface velocities are higher than the SIA ones and conversely ?

P12 L215: "at sea-level displacements of"  $\rightarrow$  this term is confusing as we are in the sea-level decrease phase. What about something like "when sea level is back to 21 and 19m above initial level".

P13 L223: "and independent of the initial conditions"  $\rightarrow$  I am not sure that this formulation is appropriate as you have actually shown that, depending on the initial state, the system might or might not go back to its initial state. I would rather say something like "The hysteresis cycles are now closed with the final steady state corresponding to the initial one".

P13 L226: It is not completely clear that the state obtained after the sea level is back to the initial 0 m level is a "viable state" (you mean a steady state, right ?), as it seems that you are keeping decreasing sea level below this initial reference.

P14 Fig.8: "The crosses represent the results of steady state branches of the transient simulations at corresponding sea levels."  $\rightarrow$  This is not very clear to me. How can you be sure that these are steady states while you keep increasing/decreasing sea-level?

P14 L253-254: "the ad hoc grounding line positions of the full Stokes model"  $\rightarrow$  I don't quite like this expression as it sounds a bit like if this GL position was fixed arbitrarily, while it is actually the most rigorous way to define it. Indeed, in the hydrostatic approximation there is the assumption that the stress in a horizontal plane is purely vertical and equals the weight of the overlying ice column (i.e. shear stresses  $t_{xz}$  and  $t_{yz}$  are neglected), which allows to deduce the GL position from the flotation criterion. In contrast, in full-stokes such assumption is not made, and a contact problem must be solved in which the normal stress at the ocean/ice interface (which is not necessarily purely hydrostatic this time) is compared to the sea water pressure.

P16 Fig. 11: 'blue lines'  $\rightarrow$  Do you mean 'solid lines' ?

P16 Fig. 11: 'after an equal increase and decrease in sea level.'  $\rightarrow$  'after a full cycle of sea level increase and decrease'

P17 Fig. 17: Panel (b) is the intermediate friction scenario and not the high friction scenario, right ?

P18 L280-282: I have a hard time to understand your point. Could you explain ?

P18 L303-304: I don't understand this point. Could you reformulate to make it clearer ?

P19 L330: 'full Stokes velocities with SIA velocities'  $\rightarrow$  'full Stokes steady velocities with SIA steady velocities' ?

P19 L331: 'due to a greater influence of stresses from the surrounding ice shelf'  $\rightarrow$  'due to stronger mechanical coupling to surrounding ice shelf' ?

**References**

- Durand G, Gagliardini O, De Fleurian B, Zwinger T and Le Meur E (2009) Marine ice sheet dynamics: Hysteresis and neutral equilibrium. Journal of Geophysical Research: Earth Surface (2003–2012), **114**(F3)
- Gagliardini O, Brondex J, Gillet-Chaulet F, Tavard L, Peyaud V and Durand G (2016) Brief communication: Impact of mesh resolution for mismip and mismip3d experiments using elmer/ice. The Cryosphere, **10**(1), 307–312

Greve R and Blatter H (2009) Dynamics of ice sheets and glaciers. Springer Science & Business Media

---

## Referee Comment (RC2)

**Review of "Hysteretic evolution of ice rises and ice rumples in response to variations in sea level" by Henry et al.**

General comments

This paper presents modelling experiments designed to test the responses of ice rises and rumples to perturbations in sea level under different basal friction conditions. It demonstrates hysteretic behaviour in velocities and profiles of ice rises, with two distinct possible steady states. Repeated tests in one case show that one of these steady states is stable, with further perturbations to sea level responding in a closed hysteretic loop. Additionally, comparisons are made between results using full Stokes model and an SIA approximation, showing minimal difference with a high friction case, but a large mismatch when greater basal sliding is allowed.

I found the majority of the manuscript to be well written and easy to follow. However, in some places the paper in its current form lacks clarity and precision. Some more attention needs to be given to the presentation of the equations in the early sections, where there are cases of conflicting notation and variables which are not defined. Part of the experimental setup described in the text is contradicted by some of the figures, namely whether the sea level is reduced back to its original state, or in fact lowered beyond the initial value. Another aspect I was unclear on is the presentation of "equilibrium states" on some figures, which did not appear to be addressed in the text (perhaps I somehow missed it, in which case it should be made clearer). Specific comments are listed below.

Overall, I found this to be an interesting study, with novel and useful results. The methodology is rigorous, and the manuscript is well structured. It is certainly worthy of publication in the Cryosphere subject to revisions which address the issues of clarity.

Specific comments

Line 21 – I think you start off with "Great progress in ice flow modelling" or similar, just to make it entirely clear right at the start.

Line 49 – Maybe you should briefly explain what a Vialov profile is. Just a short phrase such as "the solution to an idealised analytical problem".

Figure 1 – In panel (c) I would suggest labelling the cross sections as 1-3 and labelling the dome using a letter to differentiate it. I was also confused a little upon first glace by the arrow pointing to the ice dome being the same line type as the cross sections.

Line 61 – Should the gravity term be negative, since your z-axis point upwards?

Line 73 – I'd put the trace operation in non-italic font, to distinguish from variables.

Line 76 – You should probably use $u_{s,b}$ to be entirely clear.

Line 96 – M needs to be defined in the text.

Line 104 – This should be added to the reference list and cited as usual.

Line 115 – You've introduced h here when you already have $z_s$ defined. If this is a different quantity (eg. the reference plane isn't the same as z=0) this should be made clear. If not, you should be consistent with notation.

Line 118 – The standard and subscript h are the wrong way round in the first instance on this line.

Line 120 – This equation is quite confusing within the context of this paper. It's written in quite a convoluted way, and it would be far clearer if it were formulated in the same way as equation 8, especially as it is directly related by the following equation. Also, p and q are not defined.

Line 130 – The variables Q, R and $Q_R$ in this equation need to be defined in the text.

Line 132 – Same as above for L.

Figure 3 – I'm not clear about what the "equilibrium simulations" are. These don't seem to be referred to in section 2.5. More explanation is needed so that this can be understood. They are also referred to as "steady state branches". Does this mean that new experiments branch off from these points in which the sea level is kept the same until a steady state is reached?

Line 150 – The green line in Figure 3 shows this 0.02ma$^{-1}$ decrease continuing until it reaches -40m, rather than the last 2000 years being flat. I assume this is an error in the plot?

Line 152 – Why was a second cycle only done for one friction case?

Line 159 – This should probably be specified as being analysis of the *initial* steady states for clarity, since there are also different final steady states and the branches referenced in the caption of Fig. 3.

Line 160 – Is this referring to full Stokes or SIA? I assume full Stokes, but you should be clear.

Line 189 – "and" rather than "as well as".

Figure 5 – I assume these are the initial steady states? This should be stated in the caption, perhaps specifying the time (t=2000, if I'm correctly interpreting this?).

Figures 6 & 7 – Again, specify in the captions what time these velocities and cross sections are for.

Line 222-3 – I think what you're saying here is that from the final steady state obtained after one full perturbation cycle, further perturbations in sea level do not cause changes to the steady state position? Figure 8 is very clear in this regard, but I think there's probably a better way of wording it. Do you know if the same would be true of the intermediate and high friction cases?

Figure 8 – Are panels (b) and (d) really needed? I suppose the transition points are in slightly different places? It's still unclear to me what the "equilibrium states" are, as noted for Fig. 3 above.

Figure 10 – I notice the sea level goes below 0 at the end again here. I assumed it was a mistake in Fig. 3, as I don't think this decrease in sea level was mentioned in the text. Was it actually part of the experiment and if so why has it not been mentioned?

Figure 11 – I think you mean "solid" rather than "blue".

Figure 12  - Specifying the time would be nice here.

Appendix A – It seems a little odd to have a single figure with no text as an appendix. Can this figure not just be included within the main text, or is there some appendix text missing?

---

## Author Comment (AC2)

Author response to interactive comment on "Henry, A. C. J., Drews, R., Schannwell, C., and Višnjević, V.: Hysteretic evolution of ice rises and ice rumples in response to variations in sea level, EGUsphere [preprint], https://doi.org/10.5194/egusphere-2022-128, 2022."

We would like to thank the referees for the positive and thorough review of our paper. We appreciate the time taken and welcome your helpful comments. We have revised the manuscript to address your review comments (see below). Our replies are written in bold font.

**1 Reply to Anonymous Referee #1**

General comments

The paper 'Hysteretic evolution of ice rises and ice rumples in response to variations in sea level' by Clara Henry, Reinhard Drews, Clemens Schannwell and Vjeran Višnjevic is a modeling study making use of the Finite Element code Elmer/Ice in order to investigate, from synthetic threedimensional scenarios, the stability of ice rises and ice rumples, as well as the dynamical transition from one flow regime to the other depending on the amount of friction at the ice/bed interface. To this end, starting from an initial steady state corresponding to an ice rise situation, perturbation experiments consisting in cycles of sea level rise and decrease are run solving the full-Stokes set of equations. Obtained initial steady surface velocities on the grounded part are compared to their Shallow Ice Approximation (SIA) counterparts in order to quantify the importance of longitudinal stresses transmitted from the surrounding ice shelf to the grounded ice. Unsurprisingly deviations are significant when basal friction is low, whereas they become negligible in the high basal friction scenario. The transient simulations show that an increase of sea-level induces a transition from an ice rise flow regime to an ice rumple regime in all friction scenarios. However, in the high friction scenario, much higher sea level increase is required than in the other scenarios to switch from the ice rise to the ice rumple regime, and the latter is unstable (i.e. complete ungrounding rapidly occurs). Interestingly, the sea level decrease experiments bring to light a hysteretic response of grounded ice, with the grounded area and induced buttressing effect being systematically lower than in the sea level increase phases when sea level is decreased back to its initial level. Conclusions are then drawn regarding the initialisation of ice flow models as well as the inversion of basal friction parameters.

Overall, the paper is well-written, the proposed methodology is rigorous, the experiments are welldesigned, the figures are mostly clear and relevant, the supplementary video is very illustrative, and the conclusions regarding the stability of ice rises, as well as the highlighted hysteretic behavior in response to sea-level changes and associated irreversibility are significant for improvement of the accuracy of sea level rise projections. Therefore, I think the paper ought to be published and I have only a few minor modifications/comments to propose.

Response: Thank you for your comments which have resulted in a clearer presentation of our results. We have addressed the concerns regarding the grounding line implementation by running equivalent simulations using the "Discontinuous" and "Last Grounded" definitions, which show that the hysteretic behaviour also occurs for the other grounding line implementations. We have weakened the statements about the importance of model initialisation and instead emphasise the importance of perturbation history. Furthermore, we have addressed the notation inconsistencies and other minor points raised. My main point regards the logical link that is made between the hysteretic response of the ice rise to sea level rise and the requirement for careful model initialisation (e.g. 1.18 or 1338-339). I am not completely sure that this association really holds. Don't get me wrong, I totally agree that careful initialisation of models is of prime importance when running transient simulations of the future evolution of ice sheets/shelves. I also agree that "the dynamics and buttressing effect of ice rises and ice rumples are dependent on the initial geometry prescribed, which is typically unknown" (1.319-320). I see clearly the link between the hysteretic behavior and some form of irreversibility: if the system is forced with a given perturbation from a given initial steady state, it does not come back to the same steady state when the perturbation is removed. However, it does not necessarily mean it will behave dramatically differently if you start from a slightly different initial state, as long as the perturbation pattern is similar (i.e. in your case, sea level increase or sea level decrease). A good illustration of this point is the second cycle of perturbation that you impose for the low friction scenario: the initial steady state from which the system starts for this second cycle is different from the one of the first cycle, and yet the dynamical evolution of the grounded area and buttressing effect become relatively rapidly similar to that of the first cycle (dotted lines are 'rapidly' superimposed to solid lines in Figs. 8a-c). Once again, I have the feeling that it is more the history of the perturbation (are we in a sea level increase or decrease phase?) that is of importance rather than the initial state.

Response: Agreed, and thank you for raising this point. We have changed the sentence "This hysteresis not only shows irreversibility following an equal increase and subsequent decrease in sea level, but also has important implications for ice flow model initialisation." (originally line 18) to "This hysteresis not only shows irreversibility following an equal increase and subsequent decrease in sea level, but also shows that the perturbation history is important when the ice rise or ice rumple geometry is not known.". Furthermore, we have changed the sentence "As a consequence of this behaviour, we identify the need for careful consideration of the grounded area of an ice rise during model initialisation in order for the correct feature to form." to "As a consequence of this behaviour, we identify the importance of perturbation history for the formation of the correct feature." (originally lines 338-339).

Another point regards the presentation of the SIA model (Sect. 2.3 and 2.4). It seems to me that it is largely inspired from Greve and Blatter (2009), and some notations become inconsistent with the ones that were used to introduce the full-Stokes model in Sect 2.1. See specifics comments. Finally, there are a few points that, in my opinion, lack of clarity. First, I would write straight away in the abstract that you are running synthetic experiments and not dealing with real-world applications. Second, the fact that the comparison between the full-Stokes and SIA surface velocities is done for the initial steady states only would benefit to be stated more clearly in the text.

Response: Agreed. The notation inconsistencies have been rectified (see replies below). We have added to the abstract that we are studying idealised ice rises and ice rumples in the form of the sentence, "We investigate this behaviour using a three-dimensional full Stokes ice flow model with idealised ice rises and ice rumples.". We have changed the title of Section 3.1 from "Steady state analysis" to "Steady state analysis before sea level perturbation" so that it is clear that the SIA comparison is only done for the initial steady states.

Finally, although it is never clearly mentioned in the text (unless I missed something), it seems from Figs.3-8-9-10 that the sea level decrease experiments are continued after the initial 0 m level has been recovered. If this is true, this would deserve some explanation in the text.

Response: Agreed. This was not explained well. We have now added the sentence "Furthermore, we run branches of the simulation beyond the original sea level at the same sea level decrease rate of  $0.02 \text{ m a}^{-1}$ ." to Section 2.5

Below, I list some specific comments.

Specific comments

RV1.1: P2 L30: 'control' ! Is that not too strong ? What about 'influence' or 'affect' ?

Response: "control' changed to "influence'

**RV1.2:** P2 L46-47: 'simpler ice-flow approximations' ! shouldn't it be singular ?

Response: This indeed was not strictly correct. We have changed this part of the sentence to "we compare the full Stokes solutions with the shallow ice approximation (Hutter, 1983; Greve and Blatter, 2009) and the Vialov profile (Vialov, 1958)),"

**RV1.3:** P4 L62: I think there should be a minus sign in front of g as the vertical unit vector is pointing upward.

Response: Agreed. The minus sign was originally included in the table. We have now taken it out of the table and added it to the following equation:

 $\mathbf{g} = -g\hat{\mathbf{e}}_z$

**RV1.4**: P4 L64: The 0 should not be bold as the divergence of a vector is a scalar.

**Response: Fixed.**

**RV1.5:** P4 L76: Here, the vector u represents the velocities at the surface/base, which should be precised. There is an additional constraint that you did not mention and that is of importance for the grounded part, i.e.  $b(x, y) \leq zb(x, y, t) \leq zs(x, y, t)$  where b represents the bed elevation.

Response: Agreed. We have added this condition to the text.

**RV1.6:** P5 L82: How do you tune this 'tuning parameter' ?

Response: The value was deemed appropriate during initial model setup based on the fact that this particular parameter choice gave reasonable basal melt rates that allowed the formation of an ice rise. **RV1.7:** P5 L82: 'the closest' ! what do you mean by 'the closest' ?

Response: By "closest", we wanted to specify that it is not just any point on the grounding line (GL) that is used in the calculation of the distance from the grounding line, but only the GL node closest to the current node during computation. We have removed the word "closest" and added the following sentence for clarification: "During computation, x represents the position of the current node and  $x_g$  represents the position of the grounding line node closest to the current node."

**RV1.8:** P5 L85: 'no friction' ! I find more natural to speak about 'free-slip', but it is a matter of choice.

**Response: Agreed, "no friction" changed to "a free-slip condition"**

**RV1.9:** P5 L91: Why did you choose the 'First Floating' implementation ? You are dealing with Weertman friction law with constant friction coefficients, which induces a discontinuity of friction at the GL on the mathematical level. The proper way to represent this discontinuity in the FEM numerical framework of Elmer is to use the 'Discontinuous' implementation. Using the 'First Floating' implementation, you artificially impose a linear decrease of friction over all the last grounded elements due to the interpolation of friction parameters at the integration points from their nodal values through the FEM basis functions. This choice could be justified by the fact that, on the physical level, one would expect a smooth transition of friction from its value on the grounded part to zero at the GL. If that was your reasoning when choosing the FF implementation, then it would be worth to state it clearly in the text.

RV1.10: P6 L103: According to Gagliardini and others (2016), with such a resolution of 350 m, the GL behavior keeps being sensitive to the numerical treatment of friction at the GL (i.e. First Floating, Last Grounded or Discontinuous implementations). This is an additional reason to give a clear justification for your choice of the FF implementation.

Response: Thank you for raising this point. We have run an equivalent simulation using the Discontinuous grounding line implementation and have added a comparison plot in the appendix (Fig. B1 in Appendix, also shown here as Fig. 1) showing the response of the grounded area to sea level perturbation. Furthermore, we have added the following text in the appendix: "In the case of the low basal friction scenario, we have run equivalent simulations using a differing grounding line numerical implementation, namely the Discontinuous method (Fig. B1). At the grounding line, basal friction is applied if the other two nodes in the element are also grounded and a free-slip condition is applied if the other two nodes are ungrounded. The First Floating numerical implementation, however, assumes a free-slip condition at the grounding line and a linear reduction in basal friction between it and the upstream node is applied. Although the Discontinuous numerical implementation has been shown to have the least dependence on mesh resolution, it can be argued that the *First Floating* is more plausible physically, with effective pressure disappearing at the grounding line (Gagliardini et al., 2016). The simulations show that regardless of the numerical

Figure 1: Shown is the response of the grounded area in the low friction case of the *First Floating* (red) and *Discontinuous* (blue) Elmer/Ice numerical grounding line implementations.

**implementation, hysteresis occurs."**

**RV1.11:** P6 L105: It would be worth precising the order of magnitude of considered time scales here, even if it becomes clearer later in the text.

**Response: Agreed, "time scales" changed to "glacial-interglacial time scales".**

RV1.12: P6 Eq10: It is strange that you switch from zs to h and from zb to b to denote the surface/base elevations. It feels like you have taken Eq. (5.84) of Greve and Blatter (2009) without adapting the notation to your own work.

Response: Agreed. We have changed h to  $z_s$  and H to  $(z_s - z_b)$ . We have left b as is because it refers to the bed, whereas  $z_b$  refers to the bottom ice surface (i.e. the ice surface in contact with the ocean or bed).

RV1.13: P6 Eq11: Same remark as above, plus I don't understand the purpose of the subscript h for the gradient operator. First, it does not appear in Eq. (10). Second, h being the surface elevation (your zs), it does obviously not depend on z so that the gradient operator does only consist in the x and y components. In addition, in the first occurrence of the gradient operator in Eq. (11), the subscript is not at the proper position.

**Response: Agreed, we have removed the subscripts and removed the sentence, "Here, $\nabla_h$ denotes the two-dimensional, horizontal gradient operator."**

**RV1.14:** P6 Eq12: Here again, the formulation of Greve and Blatter (2009) would require some adaptations to your own case. More precisely, this expression is not directly consistent with the way the basal shear stress is defined in Eq. (8). In Greve and Blatter (2009), the effect of the normal stress Nb on the basal shear stress is explicited in the formulation of the friction law (5.35), while most of the time this effect is hidden in the friction parameter C of the Weertman law. I know that Eq. (13) is intended to establish the consistency between the two formulations, but in that case I think there should be a minus sign in Eq. (13) (or in Eq. (8), but you have to choose), and the exponents p and q that appear in Eq. (12) would need to be defined somewhere. In your case q = 1 and p = 1=m. I think it would be cleaner to have Eq. (12) formulated directly consistently with Eq. (8).

Response: We agree that there is an inconsistency in the minus signs which stems from the choice of representation of stress as either the driving force or basal drag in the stress-velocity relationship. We have changed Eq. 8 to

$$\boldsymbol{\tau}_b = -C |\mathbf{u}_b|^{m-1} \boldsymbol{u}_b, \tag{1}$$

to include a minus sign. We have also added the sentence "In Eq. (12), p and q are chosen for consistency with the non-linear Weertman-type friction law described above.", and the values are provided in the parameter table. We have not reformulated either equation as we wanted to use the common representation of each equation.

**RV1.15:** P6 Eq13: See comment above.

Response: See reply directly to RV1.14

**RV1.16:** P6 Eq17: Here, the expression of h0 does not correspond to the one in Eq. (5.117) of Greve and Blatter (2009). Could you double-check ?

Response: Yes, the expression of  $h_0$  differs by a factor of 2 in the denominator below  $\dot{a}_s$ . The reason is the use of a radial/polar flux condition compared with the standard Cartesian coordinate formulation. When Eq. (5.107) in Greve & Blatter (2009) is expressed in polar coordinates, whilst assuming no azimuthal variance, Eq. (15) is obtained. Integrating, we obtain  $Q_R(R) = \dot{a}_s R/2$  in contrast to  $Q(x) = \dot{a}_s x$  (Eq. (5.115) in Greve & Blatter (2009)). Further calculations, analogous to page 86 in Greve & Blatter (2009), result in the extra factor of 2 in the denominator in the expression of  $h_0$ .

**RV1.17:** P8 Fig. 3: The Fig. is not consistent with the main text of Sect. 2.5: the decrease of sea level seems to continue further below the initial 0 m level (green and red lines), while the text says that "Sea level is then decreased at a rate of  $0.02 \text{ m a}^{-1}$  back to the initial level followed by a second phase of constant sea level for 2000 years."

Response: Agreed. This was not explained well. We have now added the sentence

"Furthermore, we run branches of the simulation beyond the original sea level at the same sea level decrease rate of  $0.02 \text{ m a}^{-1}$ ." to Section 2.5

RV1.18: P8 L148: "with ice being frozen to the bed" ! "mimicking ice frozen to the ground" ?

**Response: Agreed. Changed to "mimicking ice frozen to the bed."**

RV1.19: P9 L162: "characteristic timescale" ! I have a hard time to figure out what does this quantity represents physically. Maybe, it requires more explanation or maybe you can simply drop it as it is not used anywhere else later on.

**Response: Agreed. We have added the sentence "The characteristic timescale is a metric that gives an indication of the rate of development of Raymond arches (Martín et al., 2009; Goel et al., 2020).".**

**RV1.20:** P9 L189-190: "the upstream ice shelf velocities" ! it is not very clear what velocities exactly you are referring to, and how you can get a single value of velocity. You give a brief explanation in the caption of Fig. 9, but some precisions would be required here too.

Response: Agreed. We have added the sentence "The upstream ice shelf velocity is defined as the mean velocity of ice in the x-direction at x = 20 km, as marked by Label (1) in Fig. 1c."

**RV1.21:** P10 Fig. 4: You should precise that the velocities are the full-Stokes ones (are they ?).

Response: Agreed. We have now specified in the caption that the velocities are the full Stokes velocities.

**RV1.22:** P11 Fig. 5: What is represented exactly ? Is is  $|v_{Stokes} - v_{SIA}|/|v_{Stokes}|$ ? If yes, wouldn't it be more interesting to represent  $(|v_{Stokes}| - |v_{SIA}|)/|v_{Stokes}|$ , so that we could tell when the full-Stokes surface velocities are higher than the SIA ones and conversely ?

Response: Agreed. We have now plotted  $100 \times (|v_{Stokes}| - |v_{SIA}|)/|v_{Stokes}|$  rather than  $100 \times |v_{Stokes} - v_{SIA}|/|v_{Stokes}|$ .

**RV1.23:** P12 L215: "at sea-level displacements of" ! this term is confusing as we are in the sea-level decrease phase. What about something like "when sea level is back to 21 and 19m above initial level".

Response: Agreed. We have replaced the text with the sentence "... when sea level is 21 and 19 m above the initial sea level in the..."

**RV1.24:** P13 L223: "and independent of the initial conditions" ! I am not sure that this formulation is appropriate as you have actually shown that, depending on the initial state, the system might or might not go back to its initial state. I would rather say something like "The hysteresis cycles are now closed with the final steady state corresponding to the initial one".

**Response: Agreed. We have changed the sentence to "The hysteresis cycle is now closed, with the final steady state corresponding to the state before the last sea level perturbation cycle."**

RV1.25: P13 L226: It is not completely clear that the state obtained after the sea level is back to the initial 0 m level is a "viable state" (you mean a steady state, right ?), as it seems that you are keeping decreasing sea level below this initial reference.

Response: Only the initial state is in steady state and the second state at 0 m is indeed a transient state. We have changed the word "viable" to "differing" so it is not interpreted as meaning "steady state".

**RV1.26:** P14 Fig.8: "The crosses represent the results of steady state branches of the transient simulations at corresponding sea levels." ! This is not very clear to me. How can you be sure that these are steady states while you keep increasing/decreasing sea-level ?

Response: What we have done is run branches of the simulation to steady state at the specified sea levels, keeping sea level fixed. For clarity, we have added the sentence "Branches of the *low* basal friction simulation are run to steady state at discrete intervals while keeping sea level fixed. We run these simulation branches in order to understand how far from steady state the transient simulations are. This gives an indication of how transient simulations with lower absolute increase and decrease rates would evolve." to Section 2.5.

RV1.27: P14 L253-254: "the ad hoc grounding line positions of the full Stokes model" ! I don't quite like this expression as it sounds a bit like if this GL position was fixed arbitrarily, while it is actually the most rigorous way to define it. Indeed, in the hydrostatic approximation there is the assumption that the stress in a horizontal plane is purely vertical and equals the weight of the overlying ice column (i.e. shear stresses txz and tyz are neglected), which allows to deduce the GL position from the flotation criterion. In contrast, in full-stokes such assumption is not made, and a contact problem must be solved in which the normal stress at the ocean/ice interface (which is not necessarily purely hydrostatic this time) is compared to the sea water pressure.

**Response: Agreed. We have removed the words "ad hoc".**

RV1.28: P16 Fig. 11: 'blue lines' ! Do you mean 'solid lines' ?

Response: Yes, we have changed "blue lines" to "solid lines".

**RV1.29:** P16 Fig. 11: 'after an equal increase and decrease in sea level.' ! 'after a full cycle of sea level increase and decrease'?

**Response: Agreed. We have changed the text to read "... after a full cycle of sea level increase and decrease.".**

**RV1.30:** P17 Fig. 17: Panel (b) is the intermediate friction scenario and not the high friction scenario, right ?

Response: Agreed. Text changed from "high" to "intermediate".

RV1.31: P18 L280-282: I have a hard time to understand your point. Could you explain?

Here we explain the counterintuitive result that surface velocities on the ice rumple are lower in the *low* friction scenario compared to *intermediate* friction scenario. The reason is that in the *low* friction scenario the total grounded area is larger compared to the *intermediate* friction scenario. It might be worth investigating whether inverse techniques used to predict the basal friction coefficient beneath pinning points reproduce these results, e.g., regardless of horizontal resolution applied. We have changed the text: 'Interestingly, the *low* friction ice rumple exhibits lower minimum velocities than the *intermediate* friction ice rumple most, likely due to a greater grounded area (Fig. 12). It is worth investigating whether inverse techniques used to predict the basal friction coefficient beneath pinning points produce results which remain valid regardless of horizontal resolution applied.' "

RV1.32: P18 L303-304: I don't understand this point. Could you reformulate to make it clearer ?

Response: Agreed, the previous formulation resulted in possible misinterpretation. We have re-worded the paragraph as, "A self-stabilising feedback occurs, with divide migration opposing grounding line retreat in a sea level increase scenario. The ice rise height reduces and the divide migrates stossward during lee side grounding line retreat. Because the divide moves stossward, the area of accumulation adjacent to the divide on the lee side of the ice rise increases. The increased accumulation area promotes an increased flux across the grounding line, opposing grounding line retreat. Analogously, sea level decrease results in leeward divide migration. The resulting reduction in accumulation area adjacent to the divide on the lee side of the ice rise opposes grounding line advance. The existence of negative feedback mechanisms in both the sea level increase and decrease scenario result in hysteretic behaviour (Figs. 8, 9 and A1)."

**RV1.33:** P19 L330: 'full Stokes velocities with SIA velocities' ! 'full Stokes steady velocities with SIA steady velocities' ?

Response: Agreed. We have changed the sentence to read "..simulated steady state full Stokes velocities with steady state SIA velocities...".

**RV1.34:** P19 L331: 'due to a greater influence of stresses from the surrounding ice shelf' ! 'due to stronger mechanical coupling to the surrounding ice shelf' ?

Response: Agreed. Text changed to "due to stronger mechanical coupling to surrounding ice shelf".

**2 Reply to Anonymous Referee #2**

This paper presents modelling experiments designed to test the responses of ice rises and rumples to perturbations in sea level under different basal friction conditions. It demonstrates hysteretic behaviour in velocities and profiles of ice rises, with two distinct possible steady states. Repeated tests in one case show that one of these steady states is stable, with further perturbations to sea level responding in a closed hysteretic loop. Additionally, comparisons are made between results using full Stokes model and an SIA approximation, showing minimal difference with a high friction case, but a large mismatch when greater basal sliding is allowed. I found the majority of the manuscript to be well written and easy to follow. However, in some places the paper in its current form lacks clarity and precision. Some more attention needs to be given to the presentation of the equations in the early sections, where there are cases of conflicting notation and variables which are not defined. Part of the experimental setup described in the text is contradicted by some of the figures, namely whether the sea level is reduced back to its original state, or in fact lowered beyond the initial value. Another aspect I was unclear on is the presentation of "equilibrium states" on some figures, which did not appear to be addressed in the text (perhaps I somehow missed it, in which case it should be made clearer). Specific comments are listed below. Overall, I found this to be an interesting study, with novel and useful results. The methodology is rigorous, and the manuscript is well structured. It is certainly worthy of publication in the Cryosphere subject to revisions which address the issues of clarity.

We would like to thank the referee for their encouraging review of our manuscript. We have addressed the issue of missing information in the model setup section describing the lowering of sea level below the original sea level in Section 2.5. We have indicated in the manuscript that we use the terms "steady state" and "equilibrium state" interchangeably and have added text to clarify how the steady states are reached (see RV1.14 below). Furthermore, we have addressed issues raised regarding the presentation of the equations and other minor points.

Specific comments

RV2.1: Line 21 – I think you start off with "Great progress in ice flow modelling" or similar, just to make it entirely clear right at the start.

Response: Agreed. We have changed the sentence accordingly.

RV2.2: Line 49 – Maybe you should briefly explain what a Vialov profile is. Just a short phrase such as "the solution to an idealised analytical problem".

**Response: Agreed. We have now added the following to the sentence: "...between the full Stokes ice thickness and the Vialov profile, an idealised solution for the ice geometry.**

RV2.3: Figure 1 – In panel (c) I would suggest labelling the cross sections as 1-3 and labelling the dome using a letter to differentiate it. I was also confused a little upon first glace by the arrow pointing to the ice dome being the same line type as the cross sections.

Response: We have altered the figure, marking the cross-sections as 1-3 and the dome with the letter "D"

**RV2.4**: Line 61 – Should the gravity term be negative, since your z-axis point upwards?

Response: Agreed. The minus sign was originally included in the table. We have now taken it out of the table and added it to the following equation:

 $\mathbf{g} = -g\hat{\mathbf{e}}_z$

RV2.5: Line 73 – I'd put the trace operation in non-italic font, to distinguish from variables.

Response: Agreed. We have now used non-italic font for the trace operator.

**RV2.6:** Line 76 – You should probably use us,b to be entirely clear.

Response: We have not made any changes here, as it is already apparent that the condition applies only to the upper and lower surface nodes and that at a numerical level, only the velocities at those nodes are used to make the calculations. The same applies, for example, to the boundary condition  $\mathbf{u} \cdot \mathbf{n} = 0$ , where we do not specify within the equation that we mean the boundary velocities.

RV2.7: Line 96 – M needs to be defined in the text.

Response: Agreed. We have added "M is the amplitude of the bed anomaly"

RV2.8: Line 104 – This should be added to the reference list and cited as usual.

Response: Agreed. Cited in text and added to reference list.

RV2.9: Line 115 – You've introduced h here when you already have zs defined. If this is a different quantity (eg. the reference plane isn't the same as z=0) this should be made clear. If not, you should be consistent with notation.

Response: Agreed. We have replaced h with  $z_s$  and H with  $(z_s - z_b)$ . See also reply to *RV1.12*.

RV2.10: Line 118 – The standard and subscript h are the wrong way round in the first instance on this line.

**Response: Agreed. We have removed the subscripts completely as suggested by reviewer one.**

RV2.11: Line 120 – This equation is quite confusing within the context of this paper. It's written in quite a convoluted way, and it would be far clearer if it were formulated in the same way as equation 8, especially as it is directly related by the following equation. Also, p and q are not defined.

Response: We have decided to keep the differing formulations for the FS and SIA basal friction parameterisations as the respective formulations are commonly written in these forms in the literature for Elmer/Ice and SIA. We have added the sentence, "In Eq. (12), p and q are chosen for consistency with the non-linear Weertman-type friction law described above."

RV2.12: Line 130 – The variables Q, R and QR in this equation need to be defined in the text.

**Response: Agreed. We have added definitions to the text.**

RV2.13: Line 132 – Same as above for L.

Response: Agreed. We have added "L is the horizontal distance from the ice rise divide to the grounding line"

RV2.14: Figure 3 – I'm not clear about what the "equilibrium simulations" are. These don't seem to be referred to in section 2.5. More explanation is needed so that this can be understood. They are also referred to as "steady state branches". Does this mean that new experiments branch off from these points in which the sea level is kept the same until a steady state is reached?

Response: Yes, by equilibrium simulation, we mean steady state simulation. We have added the term to Section 2.5 in the sentence, "Branches of the *low* basal friction simulation are run to steady state (equilibrium) at discrete intervals while keeping sea level fixed."

*RV2.15:* Line 150 – The green line in Figure 3 shows this  $0.02\text{ma}^{-1}$  decrease continuing until it reaches -40m, rather than the last 2000 years being flat. I assume this is an error in the plot?

Response: This is not an error in the plot. We had not explained this previously, but we have now added the sentence "Furthermore, we run branches of the simulation beyond the original sea level at the same sea level decrease rate of 0.02 m a-1." to Section 2.5.

RV2.16: Line 152 – Why was a second cycle only done for one friction case?

Response: A second cycle was performed only on one friction case because of computation time. One sea level perturbation cycle took roughly 4 weeks to simulate.

RV2.17: Line 159 – This should probably be specified as being analysis of the initial steady states for clarity, since there are also different final steady states and the branches referenced in the caption of Fig. 3.

Response: Agreed. We have changed the name of section 3.1 from "Steady state analysis" to "Steady state analysis before sea level perturbation"

RV2.18: Line 160 – Is this referring to full Stokes or SIA? I assume full Stokes, but you should be clear.

**Response: we have added "full Stokes" to the sentence.**

RV2.19: Line 189 – "and" rather than "as well as".

**Response: Agreed and changed.**

*RV2.20:* Figure 5 – I assume these are the initial steady states? This should be stated in the caption, perhaps specifying the time (t=2000, if I'm correctly interpreting this?).

**Response: Agreed. We have added time to the caption and have also stated that the figures correspond to steady states.**

RV2.21: Figures 6 & 7 – Again, specify in the captions what time these velocities and cross sections are for.

**Response: Agreed. Time added to captions.**

RV2.22: Line 222-3 – I think what you're saying here is that from the final steady state obtained after one full perturbation cycle, further perturbations in sea level do not cause changes to the steady state position? Figure 8 is very clear in this regard, but I think there's probably a better way of wording it. Do you know if the same would be true of the intermediate and high friction cases?

**Response: Agreed and rephrased from "The hysteresis cycles are now closed and independent of the initial conditions." to "The hysteresis cycle is now closed, with the final steady state corresponding to the state before the last sea level perturbation cycle."**

RV2.23: Figure 8 – Are panels (b) and (d) really needed? I suppose the transition points are in slightly different places? It's still unclear to me what the "equilibrium states" are, as noted for Fig. 3 above.

Response: We decided to keep the panels (b) and (d) as the dashed line is covered by the solid lines in (a) and (c) in a large part of the plot.

RV2.24: Figure 10 – I notice the sea level goes below 0 at the end again here. I assumed it was a mistake in Fig. 3, as I don't think this decrease in sea level was mentioned in the text. Was it actually part of the experiment and if so why has it not been mentioned?

Response: We have added the sentence "Furthermore, we run branches of the simulation beyond the original sea level at the same sea level decrease rate of 0.02 m a-1." in Section 2.5

**RV2.25:** Figure 11 – I think you mean "solid" rather than "blue".

**Response: Agreed and changed.**

**RV2.26:** Figure 12 - Specifying the time would be nice here.

**Response: Agreed. Time added to caption.**

RV2.27: Appendix A – It seems a little odd to have a single figure with no text as an appendix. Can this figure not just be included within the main text, or is there some appendix text missing?

Response: Thank you for raising this point. We have added text describing the figure and comparing it with the low basal friction scenario.

**References**

- O. Gagliardini, J. Brondex, F. Gillet-Chaulet, L. Tavard, V. Peyaud, and G. Durand. Brief communication: Impact of mesh resolution for MISMIP and MISMIP3d experiments using Elmer/Ice. *The Cryosphere*, 10(1):307–312, Feb. 2016. doi: 10.5194/tc-10-307-2016.

[revised manuscript text omitted]
 \left( \boldsymbol{\tau} - \underline{\boldsymbol{p}} \underline{\boldsymbol{\mathcal{P}}} \mathbf{I} \right) + \rho_i \mathbf{g} = 0, \tag{1}$$

where  $\tau$  is the deviatoric stress tensor, p - p is the pressure,  $\rho_i$  is the ice density and  $\mathbf{g} = g\hat{\mathbf{e}}_z \cdot \mathbf{g} = -g\hat{\mathbf{e}}_z$  is the gravitational 65 acceleration. We assume the ice to be incompressible, and so, the mass conservation equation reduces to

$$\nabla \cdot \mathbf{u} = 0. \tag{2}$$

The non-linear rheology of ice is modelled using Glen's flow law, which relates the deviatoric stress tensor,  $\tau$ , to the strain rate tensor,  $\dot{\epsilon}$ , as

$$\boldsymbol{\tau} = 2\eta \boldsymbol{\dot{\epsilon}},\tag{3}$$

70 where the effective viscosity,  $\eta$ , is

$$\eta = \frac{1}{2} A^{-1/n} \dot{\epsilon}_e^{(1-n)/n}.$$
(4)

Here, n is the Glen's flow law exponent, A is a rheological parameter primarily dependent on ice temperature. Since we assume ice to be isothermal, A is set to a constant value in all simulations. The effective strain rate,  $\dot{\epsilon}_e$ , is calculated from the strain rate tensor,  $\dot{\epsilon}$ , as

75  $\dot{\epsilon}_e = \sqrt{tr(\dot{\epsilon}^2)/2}\sqrt{tr(\dot{\epsilon}^2)/2}.$ (5)

**2.1.1 Boundary conditions**

80

The upper surface,  $z = z_s(x, y, t)$ , and lower surface,  $z = z_b(x, y, t)$ , evolve subject to

$$\left(\frac{\partial}{\partial t} + \mathbf{u} \cdot \boldsymbol{\nabla}\right)(z - z_{s,b}) = \dot{a}_{s,b},\tag{6}$$

where  $\dot{a}_{s,b}$  are the accumulation / melt rates at the ice-shelf surface (s) and ice shelf base (b), respectively.

Furthermore, the grounded portion is constrained by the condition

$$\underline{b}(x,y) \le \underline{z}_b(x,y,t) \le \underline{z}_s(x,y,t),\tag{7}$$

---

## Editor Decision (ED1)

**Final technical corrections for egusphere-2022-128**

- remove double parenthesis in line 49.

- equation (6) : instead of $\mathring{a}_{s,b}$, it would make more sense to write the sum of surface mass balance and basal mass balance ($\mathring{a}_s+\mathring{a}_b$) which are defined a few lines below.

- as equation (8) includes 1/50 and 1/100 (which have units), it is important to specify the units of $H$, **x**, $\mathbf{x_g}$ and $\mathring{a}_b$.